# Genetic parallelism underpins convergent mimicry coloration in Lepidoptera across 120 million years of evolution

Yacine Ben Chehida[1,2�و], Eva S. M. van der Heijden[3,4�و], Edward Page[1�],
Patricio A. Salazar C[1,3,4], Neil Rosser[5,6], Kimberly Gabriela Gavilanes Córdova[7],
Mónica Sánchez-Prado[7], María José Sánchez-Carvajal[7], Franz Chandi[7],
Alex P. Arias-Cruz[7], Maya Radford[1], Gerardo Lamas[8], Chris D. Jiggins[4], James Mallet[6],
Melanie McClure[9], Camilo Salazar[10], Marianne Elias[11,12,13], Caroline N. Bacquet[7],
Nicola J. Nadeau[2], Kanchon K. Dasmahapatra[1*], Joana I. Meier[3,4]

1 Leverhulme Centre for Anthropocene Biodiversity, Department of Biology, University of York, York,
United Kingdom, 2 Ecology and Evolutionary Biology, School of Biosciences, University of Sheffield,
Sheffield, United Kingdom, 3 Tree of Life Programme, Wellcome Sanger Institute, Cambridge, United
Kingdom, 4 Department of Zoology, University of Cambridge, Cambridge, United Kingdom, 5 Department
of Biology, University of Miami, Coral Gables, Florida, United States of America, 6 Department of
Organismic and Evolutionary Biology, Harvard University, Cambridge, Massachusetts, United States
of America, 7 Universidad Regional Amazónica IKIAM, Tena, Ecuador, 8 Museo de Historia Natural,
Universidad Nacional Mayor de San Marcos, Lima, Peru, 9 Laboratoire Écologie, Évolution, Interactions
des Systèmes Amazoniens (LEEISA), Université de Guyane, CNRS, IFREMER, Cayenne, France,
10 Department of Biology, School of Sciences and Engineering, Universidad del Rosario, Bogotá,
Colombia, 11 Institut Systématique, Evolution, Biodiversité, MNHN-CNRS-EPHE-SU-UA, Muséum
National d'Histoire Naturelle, Paris, France, 12 Centre Interdisciplinaire de Recherche en Biologie, Collège
de France-CNRS-INSERM-PSL, Paris, France, 13 Smithsonian Tropical Research Institute, Panama City,
Panama

و These authors contributed equally to this work.
* kanchon.dasmahapatra@york.ac.uk

journal.pbio.3003742

University, UNITED STATES OF AMERICA

**Peer Review History:** PLOS recognizes the
benefits of transparency in the peer review
process; therefore, we enable the publication
of all of the content of peer review and
author responses alongside final, published
articles. The editorial history of this article is
available here: https://doi.org/10.1371/journal.
pbio.3003742

## Abstract

Convergent evolution, the repeated evolution of similar phenotypes, is widespread
in nature, but there are few studies investigating the genetic mechanisms of conver-
gence across wide evolutionary timescales. The extent to which the same genetic
mechanisms contribute to convergent evolution could reveal whether the pathway
towards these fitness optima is flexible or constrained to follow a particular route,
informing us about the predictability of evolution. Wing color pattern mimicry in
Lepidoptera is a well-known example of convergent evolution, but as studies are
restricted to a few closely related species, it is difficult to make general inferences
about the predictability of evolution in this system. Here we study convergent evo-
lution in multiple mimetic neotropical lepidopteran lineages that diverged between
~1 and 120 Mya, including seven species of Ithomiini and *Heliconius* butterflies and
a day-flying *Chetone* moth. Across butterfly lineages that diverged up to ~30 Mya,
the genetic variants most strongly associated with convergent color pattern switches
are located in similar noncoding regions near the genes *ivory* and *optix*. In the more
distantly related moth species, color pattern variation is associated with a ~1 Mb

**Data availability statement:** All raw sequencing data are available via the European Nucleotide Archive under BioProjects PRJNA1425777, PRJEB63068, and PRJEB87706. Sample accessions and wing phenotypes are provided in S1 Table. The *Chetone histrio* and *Hypothytis anastasia* reference genomes have been deposited at DDBJ/ENA/GenBank under the accessions JBVTKZ000000000 and JBVTLC000000000, respectively. Data underlying analyses and figures are available along with code to reproduce these analyses at https://zenodo.org/records/19135682 and https://github.com/yacinebenchehida/Ithominii_convergence/tree/main.

**Funding:** This work was funded by NERC grant NE/T008121/1 awarded to K.K.D. (https://www.ukri.org/councils/nerc/); a Wellcome Trust award 220540/Z/20/A (https://wellcome.org/), a Branco Weiss Fellowship (https://brancoweiss-fellowship.org/), a Royal Society University Research Fellowship (URF\R1\221041; https://royalsociety.org/) and a Bateson Research Fellowship by St John's College, Cambridge (https://www.joh.cam.ac.uk) awarded to J.I.M.; a NERC DTP C-CLEAR PhD studentship (https://nercdtp.esc.cam.ac.uk/programme) and funding by the Zoology Department of the University of Cambridge, and St. John's College, Cambridge to E.v.d.H.; a NERC ACCE DTP PhD studentship to E.P. (https://accedtp.ac.uk/); an ANR grant (SPECREP - ANR-14-CE02-0011-01; https://anr.fr/) awarded to M.E. The funders had no role in study design, data collection and analysis, decision to publish, or preparation of the manuscript.

**Competing interests:** The authors have declared that no competing interests exist.

**Abbreviations:** ANLA, Autoridad Nacional de Licencias Ambientales of Colombia; GQ, genotype quality; GWA, genome-wide association; ML, maximum likelihood; NJ, Neighbor-Joining; PBS, phosphate-buffered saline; PCA, Principal Component Analysis; QTL, quantitative trait locus; QUAL, quality score; sgRNA, single guide RNA; SNPs, single-nucleotide polymorphisms; TF, transcription factor; WGA, Wheat Germ Agglutinin.

inversion which also contains *ivory*, closely mirroring the supergene architecture of the co-mimetic butterfly *Heliconius numata*. In contrast to previous studies on *Heliconius* butterflies, we find limited evidence that convergence among closely related Ithomiini species results from alleles shared by hybridization. Repeated parallel evolution of regulatory switches via reuse of the same two genes suggests that convergent color pattern evolution is highly constrained and predictable even across large evolutionary timescales. Such constraints may have facilitated diverse taxa joining this species-rich mimicry ring.

Convergent or parallel evolution is a natural experiment where unrelated species independently evolve similar traits in response to similar selective pressures. It informs us about the extent to which evolution is repeatable and thus predictable [1–4]. Highly divergent lineages can show strong trait convergence, for example, associated with the repeated colonization of land, water, or air [5] or the repeated evolution of resistance to challenges like insecticides or drought and heat stress [6,7].

Trait convergence in different species can be caused by genetic changes at different genes or the same gene ("gene reuse"). Gene reuse is predicted to be more common among closely related lineages or when developmental pathways towards shared fitness optima are constrained [8,9]. Where genes are reused, convergence may result from independent mutations at the same gene or because the same alleles are reused ("allele sharing"), either from ancestral standing variation [10,11], or as a result of introgression between species [9,12]. Allele sharing is expected mainly among very closely related species [13,14]. Examples demonstrating convergent evolution through each of these mechanisms are known, but few systems exist where the same phenotypes have evolved multiple times in highly replicated responses to the same selective pressures. Here we investigate the genetic basis of trait convergence in a system comprising lineages diverging ~1–120 Mya.

Mimicry rings are spectacular examples of convergent evolution in which multiple, distantly related, sympatric taxa converge on the same phenotype, usually in the context of aposematic signaling [15–18]. Most of our knowledge about the genetics of convergence in mimicry rings stems from studies on single butterfly species or genera, where discrete mimetic phenotypes are found within the same species [19–25]. In four species of *Papilio* butterflies, female-limited Batesian mimicry (where palatable species mimic toxic species) is controlled by reuse of the gene *doublesex* [19]. In *Heliconius* butterflies, where the mimicry is Müllerian (all mimetic species are unpalatable and share the cost of educating predators), convergent color patterning between co-mimetic subspecies of *Heliconius erato* and *Hel. melpomene* (~10 MY divergent; Fig 1) mostly results from the reuse of the *optix*, *ivory* (formerly attributed to *cortex*), and *WntA* genes via independent mutations at regulatory regions [20,21]. Mimicry among very closely related *Heliconius* species often results from allele sharing via introgression at these genes [22–24]. Structural variants such as inversions can maintain tightly linked groups of loci, preventing recombination and the

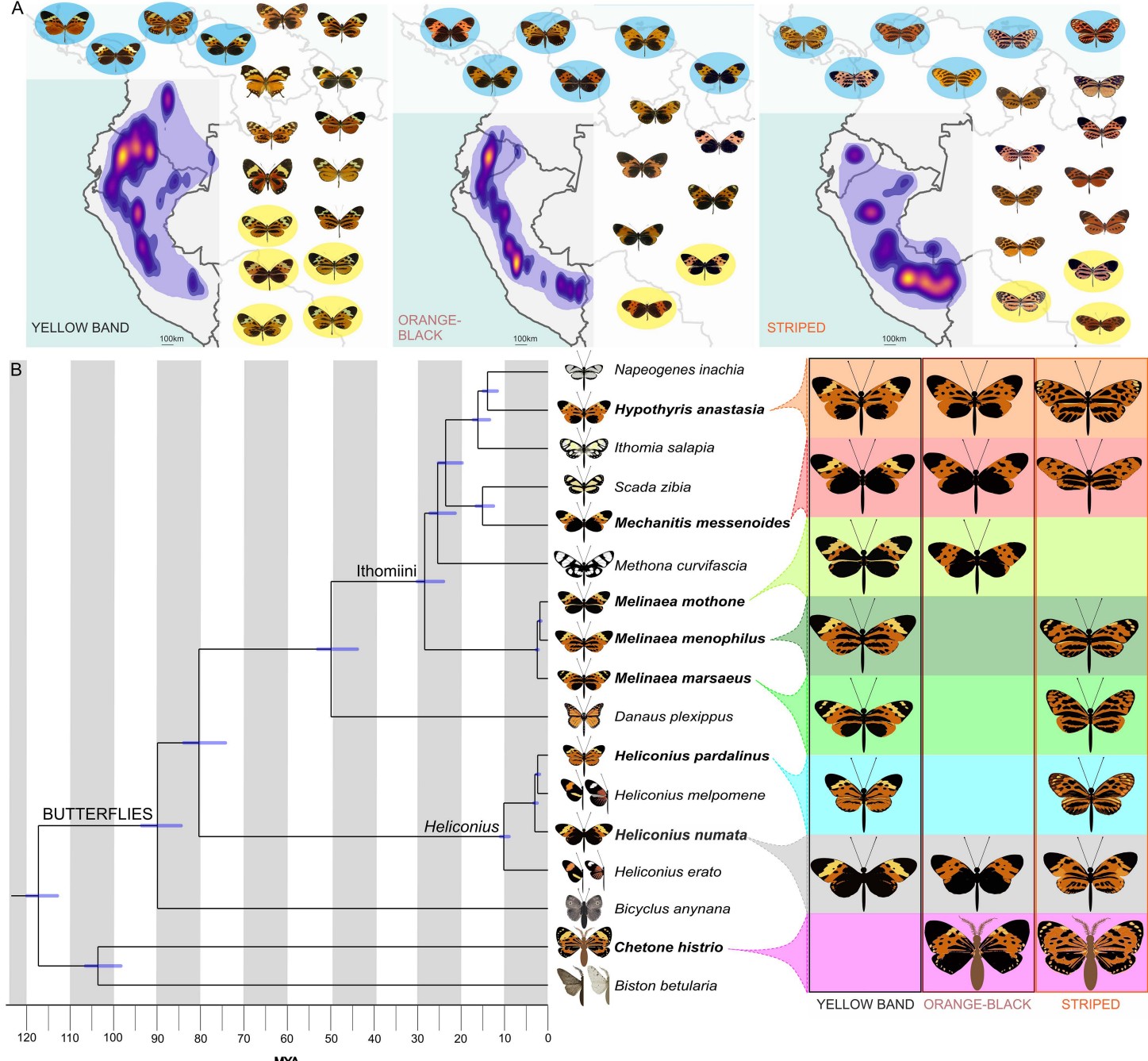

**Fig 1. Phylogenetic relationships and mimetic phenotypes. (A)** Geographic distributions of the three tiger sub-mimicry rings: yellow band, orange-black and striped, based on Doré and colleagues 2021 [26] together with representative mimicry ring member taxa (detailed in S1 Fig). The purple outline shows the maximum range of each sub-mimicry ring with the approximate abundance of taxa indicated via heatmap colouration. Taxa with a blue background were investigated using GWA/QTL. Additional taxa used in gene flow analyses are shown on a yellow background. Maps were generated using the "Stamen terrain lines" (https://maps.stamen.com/terrain-lines/) base map openly available through Stadia Maps (stadiamaps.com). **(B)** Time-calibrated phylogeny including in bold the eight lepidopteran species investigated in detail. Blue bars represent 95% confidence intervals of the node ages. Large images depict the phenotypic variation of the species investigated in this study. Smaller images also show some representative species that do not form part of the tiger mimicry ring.

production of nonmimetic phenotypes, as found in *Hel. numata* where color pattern differences between distinct mimetic phenotypes are controlled by multiple overlapping inversions containing *ivory* [25].

The neotropical "tiger" mimicry ring to which *Hel. numata* and some other heliconiine species belong is exceptionally species-rich, including over 100 species from five lepidopteran families (Figs 1 and S1). It is dominated by many chemically-defended ithomiine species, and also includes day-flying moths among other taxa (Fig 1; [26,27]). Many of the species exhibit color pattern variation, where two or more subspecies are members of different sub-mimicry rings: orange-black, yellow band, or striped (Fig 1). The genetic basis of their mimicry is unknown. Here we use this replicated natural experiment in convergent evolution, where mimetic lineages have diverged between ~1 and 120 million years ago, to i) test how divergence time shapes gene reuse during repeated adaptation [9] and; ii) where the same genes are reused, to test the contribution of introgressive allele sharing. Specifically, we use within-species genome-wide association (GWA) analyses to understand the genetic architecture of three mimetic phenotypes, that differ in the presence/absence of the forewing yellow band and the extent of hindwing melanisation, in seven species from five genera (Fig 1).

## Forewing yellow band in Ithomiini butterflies: repeated reuse of *ivory*

Using whole-genome sequences of 285 wild-caught individuals, GWA was used to find genotypic associations with the presence or absence of the forewing yellow band in the ithomiine species *Melinaea mothone* (49 specimens), *Melinaea menophilus* (64 specimens), *Mechanitis messenoides* (111 specimens), and *Hypothyris anastasia* (61 specimens). In all comparisons, clusters of significantly associated SNPs were identified in the long noncoding RNA *ivory* (near the gene *cortex*) which also controls melanisation patterns in other Lepidoptera (Figs 2 and S2; [29,31–35]). In all cases, alleles associated with the presence of the yellow band are recessive (S2–S6 Figs). A small number of SNPs in narrow 1,155–2,140 bp genomic intervals are perfectly associated with the phenotype (Figs 2 and S2–S6) in all species apart from *Melinaea menophilus,* where the peak of association is relatively broad (7,373 bp). We find surprising concordance in the location of the genomic intervals controlling an identical mimetic phenotype across these four species (Figs 2 and 3). While there is no clear sequence homology in the identified regions, in all four species they lie within the first intron of *ivory*, 25,800−33,500 bp downstream of the *ivory* promoter and a short distance upstream of the conserved E230 element, an *ivory* cis-regulatory region in the butterfly *Junonia coenia* [29]. Tian and colleagues [34] demonstrated that the microRNA *mir-193* derived from *ivory* is likely the main effector gene, repressing multiple pigmentation genes. The concordant GWA peaks likely indicate conservation of transcriptional control of *ivory* over ~28 million years of evolution in the Ithomiini.

## Hindwing melanisation in Ithomiini butterflies: repeated reuse of *optix*

We next used GWA to uncover genotypic associations with variation in black vs orange hindwing patterning in *Mel. marsaeus*, *Mel. menophilus*, *Mec. messenoides* and *Hyp. anastasia*. In all comparisons, we found peaks of association upstream of the known color patterning gene *optix*, with genotype-phenotype correlations ranging from 0.54 to 1 (Fig 2). In most cases, these were the SNPs with the strongest associations across the genomes (S2 Fig). Additional associated SNPs in some species are likely a result of population structure correlated with the phenotypes. The regions of peak association in the two *Melinaea* species correspond closely, and they are near but not overlapping the region identified in *Hyp. anastasia*. All three genomic intervals fall within the wider associated region identified in *Mec. messenoides* (Fig 3).

In *Mec. messenoides* we additionally investigated genetic associations with black vs orange patterning in the forewing base and tip, which also yielded associated SNPs near *optix*. The SNPs showing the strongest genotype-phenotype correlations for the black vs orange patterns in different wing regions fall in three adjacent genomic regions (Figs 4 and S10–S13). These may correspond to separate cis-regulatory modules of *optix* controlling different aspects of wing melanisation, similar to those proposed for different red/black phenotypes in *Heliconius* [24,36]. The two regions of association we find upstream of *optix* in *Mel. marsaeus* (Fig 2) may also represent separate cis-regulatory modules, but our samples

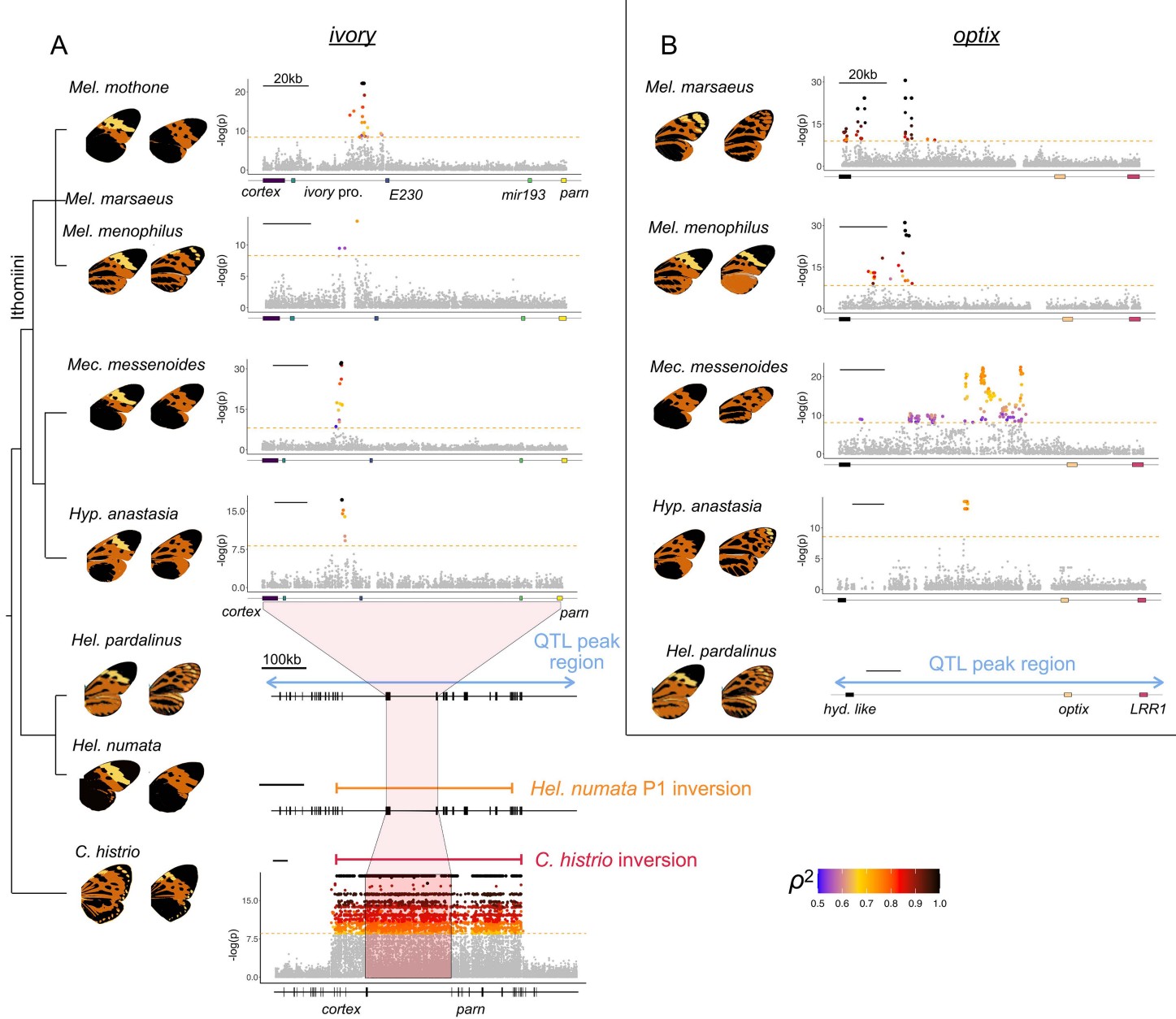

**Fig 2. *Ivory* and *optix* control convergent phenotypes across multiple species.** Zoomed Manhattan plots of genome-wide association for wing pattern variation involving **(A)** *ivory* and **(B)** *optix*. SNPs above the Bonferroni-corrected significance threshold (orange dashed line) are colored according to the strength of correlation ($\rho^2$, squared Spearman's rank correlation coefficient) between genotype and the phenotypes compared as shown in the wing images. Black points represent SNPs fully associated with the phenotypes. The bottom Manhattan plot shows the wider ~1 Mb region of high association in *Chetone histrio* around the *ivory* region. The results for *Heliconius numata* were retrieved from [28]. *Heliconius pardalinus* results are based on QTL mapping (S16 Fig). E230: *cis*-regulatory element 230 [29]; *ivory* pro.: *ivory* promoter; *hyd. like*: hydrolyze like; *LRR1*: Leucine Rich Repeat Protein 1. Association plots across the whole-genome are shown in S2 Fig. The ~400 kb P1 inversion (orange segment) is associated with color pattern variation in *Heliconius numata* [28,30] and corresponds closely in location to the *Chetone histrio* inversion as shown by the red segment (S24 Fig).

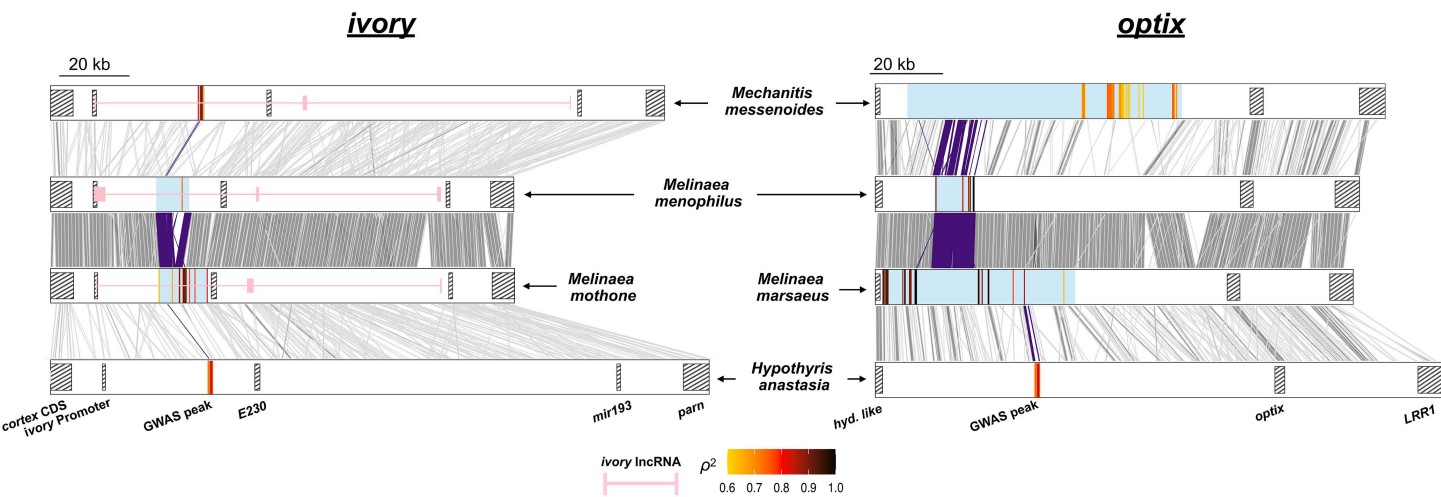

**Fig 3. Concordant locations of SNPs associated with convergent phenotypes at *ivory* and *optix*.** Blue-shaded blocks represent intervals containing SNPs significantly associated with convergent wing phenotypes (Fig 2). Homologous regions (1,000 bp segments) are shown with gray lines between species pairs. Purple lines connect regions of homology within GWA intervals that are shared across adjacent species pairs, highlighting conserved areas. Within each species, the SNPs most strongly associated with the phenotype are colored according to the strength of correlation with the phenotype ($\rho^2$, squared Spearman's rank correlation coefficient). Partial *ivory* annotations are shown in pink for three of the species. Hatched boxes mark annotated genomic features: E230: *cis*-regulatory element 230 [29]; *hyd. like*: hydrolyze like; *LRR1*: Leucine Rich Repeat Protein 1.

lack the phenotypic variation required to test this hypothesis. In contrast, black vs orange patterning in the forewing tip of *Mel. menophilus* was strongly associated (genotype-phenotype correlations of 0.87) with SNPs located between the *Hox* genes *Antennapedia* (*Antp*) and *Ultrabithorax* (*Ubx*) (S14, S15 Figs). As *Ubx* expression in butterflies is restricted to the hindwing, this phenotype likely arises through modulation of *Antp* [37]. This is similar to *Bicyclus anynana*, where *Antp* and *Ubx* promote eyespot development in fore- and hind-wings respectively [38].

## Genetic architecture of color patterning in *Heliconius pardalinus*

While the genetic basis of mimicry ring switches is known for multiple *Heliconius* species, the *Heliconius* species joining the tigerwing Ithomiini-dominated mimicry rings have received relatively little attention, with the exception of *Hel. numata* [25,28,39]. We thus investigated the loci controlling color patterning in *Hel. pardalinus* by mapping variation segregating in 82 backcross individuals between subspecies belonging to different tiger sub-mimicry rings. The only significant quantitative trait locus (QTL) for forewing yellow patterning is on chromosome 15 and contains *ivory* (Figs 2 and S16). A QTL for forewing orange patterning is found on chromosome 18, and QTLs for hindwing orange patterning are found on chromosomes 13 [40] and 18 (Figs 2 and S16). Both of the chromosome 18 QTLs encompass *optix*. These results are consistent with our findings from Ithomiini, showing that both *ivory* and *optix* predictably control convergent phenotypes across 80 My of divergence time.

## Genetic architecture of color patterning in the moth *Chetone histrio*

Unlike most tiger mimicry ring species in which different mimetic forms within-species have broadly parapatric distributions (S17–S22 Figs), the moth *Chetone histrio* is locally polymorphic in Peru with individuals belonging to either the striped (*C. histrio histrio*) or orange-black (*C. histrio hydra*) sub-mimicry rings, which consistently differ in multiple color pattern elements across both fore- and hind-wings (S23 Fig), akin to the similarly locally polymorphic species *Heliconius numata*. Comparing striped and orange-black *C. histrio*, we again find a GWA peak around *ivory*. However, unlike in the previous GWA results, the *C. histrio* GWA shows a ~1 Mb block of SNPs in perfect association with the phenotype (Fig 2). This genomic interval includes

 

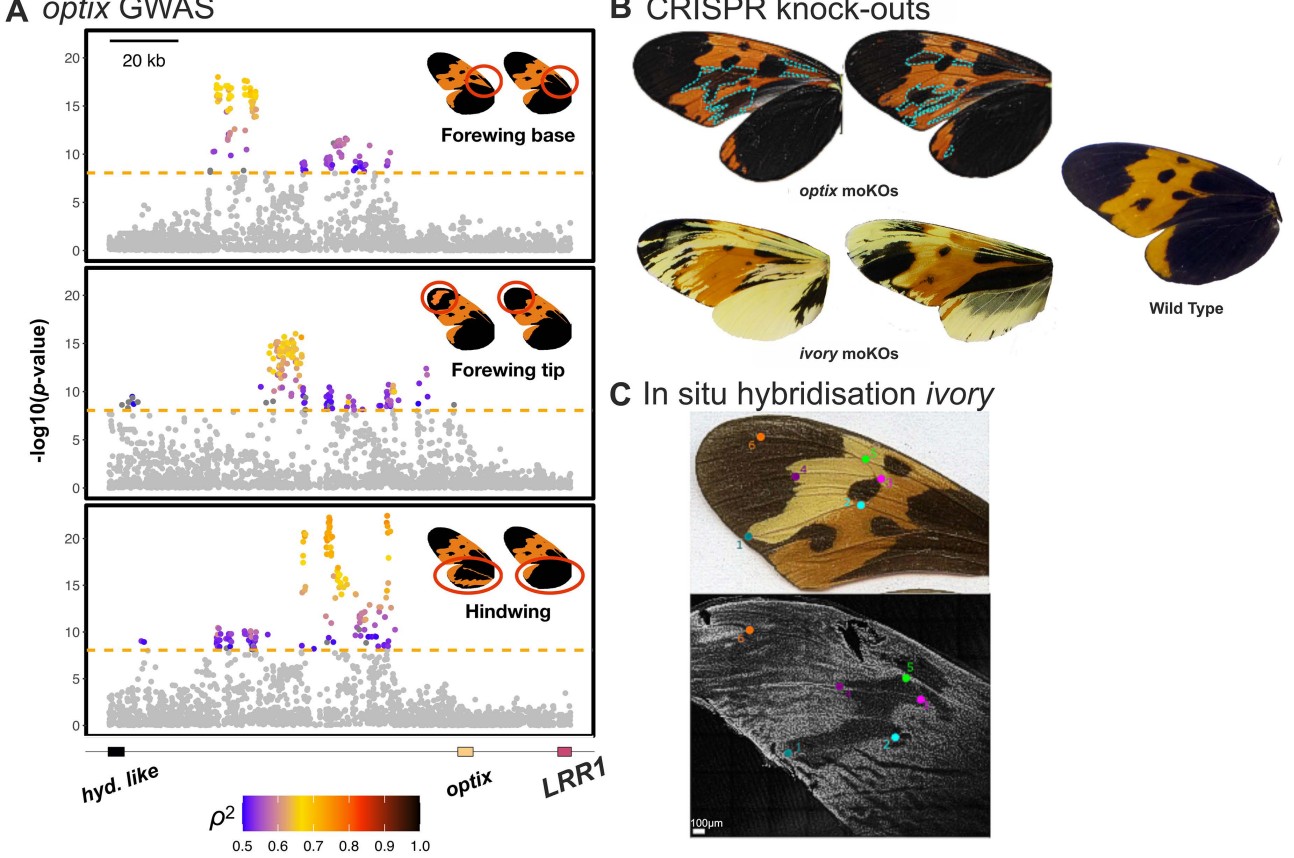

**Fig 4. Detailed phenotypic and functional analysis in *Mechanitis messenoides*. (A)** Genome-wide association analysis for separate forewing base, forewing tip and hindwing black vs orange phenotypes (red-circled regions on wings) indicate three distinct *cis*-regulatory modules of *optix*. SNPs above the significance threshold (dashed orange line) are colored according to the strength of association between genotype and phenotype (squared Spearman's rank correlation coefficient, ρ²). *hyd. like* (hydrolase-like) and *LRR1* (leucine rich repeat protein 1). Genome-wide association plots are shown in S10 Fig. **(B)** CRISPR mutagenesis of *optix* and *ivory* in *Mec. messenoides*. Wings of mosaic knockout individuals are shown: orange scales turn black in *optix* mutants (blue dashed lines highlighting mutant patches), orange and black scales turn yellow in *ivory* mutants. Additional mutants are shown in S33 Fig. **(C)** HCR in situ hybridization for *ivory* in *Mec. messenoides messenoides*. The top shows an adult forewing, and the bottom, a day 3 pupal forewing stained for *ivory*. Regions of *ivory* expression appear as white staining. The colored dots indicate vein-based wing landmarks demarcating the yellow band region which lacks *ivory* expression. S34 Fig shows *Mec. messenoides deceptus*.

*ivory* and several other flanking genes. Examination of Illumina read-pair orientation and mapped insert sizes demonstrates that this block corresponds to a 1.018 Mb inversion (S24 Fig). The striped *C. histrio histrio* are homozygous for the ancestral gene order, and the orange-black *C. histrio hydra* are either heterozygous or homozygous for the derived inversion.

Tiger color pattern polymorphism in *Hel. numata* is also maintained via an inversion architecture, and in both species the inversions likely maintain allelic combinations ("supergenes") that simultaneously control multiple color pattern elements [25]. One of the breakpoints of the *C. histrio* inversion closely matches the location of the *Hel. numata* P1 inversion, falling between the sucrose-6-phosphate hydrolase and glutaminyl-peptide cyclotransferase genes [28]. The other inversion breakpoint of *C. histrio* is located within eight genes of that of *Hel. numata* (S24 Fig). Similar to *C. histrio*, the derived P1 inversion in *Hel. numata* is dominant and controls black vs orange patterning. The similarities in genomic architecture between *C. histrio* and *Hel. numata* demonstrate striking parallel evolution not only in gene usage but also in genetic architecture, location and dominance of the inversion haplotypes between lineages that diverged ~120 MYA.

PLOS Biology

## Functional characterization in *Mechanitis messenoides*

To confirm that the genes nearest to the GWA peaks, *ivory* and *optix*, are causally involved in color pattern formation in ithomiine butterflies, we performed CRISPR-Cas9 gene knock-outs in *Mec. messenoides*. In mosaic *ivory* knockout individuals, black and orange scales turn yellow, and *optix*-knockout causes orange scales to turn black (Figs 4B and S33), consistent with findings in other butterflies [29,41,42]. In situ hybridization of pupal forewing discs (50 hours post pupation) using *ivory* probes demonstrates that while *ivory* expression occurs across the entire forewing in the orange-black *Mec. messenoides deceptus*, in the yellow-banded *Mec. messenoides messenoides*, a region lacking *ivory* expression prefigures the adult forewing yellow band phenotype (Figs 4C and S34). In contrast, staining with antibodies against *cortex*, a gene overlapping with *ivory* that has been reported to affect color patterns in Lepidoptera [35], shows no spatial association with the forewing yellow band phenotype, further supporting the role of *ivory* rather than *cortex* in wing melanisation as recently reported for other butterflies [42] (S35 Fig). Comparing yellow-banded with nonyellow-banded individuals, we do not find that *ivory* is differentially expressed in forewing pupal wing discs of *Mec. messenoides* or *Mel. menophilus* (S36 Fig). However, this is not unexpected as whole forewing tissues were used and the anticipated difference in *ivory* expression is ~20% (proportional to the area of the yellow band region compared to the whole wing) and thus not detectable in these experiments.

In *Mec. messenoides*, only 5 SNPs located in a ~1.5 kb interval (CHROM6:6,877,302−6,878,798) are fully associated with the yellow band phenotype. These are candidate binding sites for transcription factors controlling *ivory* expression. In this interval, we find eight sequence motifs which contain a fixed SNP and are present in all homozygotes of one subspecies and absent in all homozygotes of the other subspecies. Of these, four correspond closely to binding sites of known transcription factors that are also expressed in pupal wing discs of *Mec. messenoides* (Sox15, Ftz-F1, ttk, and br-Z4; S37 Fig). Sox15 is potentially implicated in the differentiation of lepidopteran scale cells [43]. In *Junonia* butterflies, Ftz-F1 has been shown to bind to the *ivory* promoter [29]. These SNPs and transcription factors are strong candidates for future investigation of transcriptional control of *ivory*.

## Limited evidence for introgressive allele sharing

Convergent evolution among very closely related *Heliconius* species often results from the sharing of adaptive alleles among species via occasional hybridization [22–24]. Both the forewing yellow band and the hindwing black mimetic phenotypes are found in multiple *Melinaea* and *Hypothyris* species (and to a limited extent in *Mechanitis*) allowing us to test if allele sharing via introgression facilitated color pattern convergence. In addition to the species used in the GWAs, we analyzed the genomes of 222 individuals of 24 species (8 *Melinaea*, 9 *Hypothyris*, and 7 *Mechanitis* species; S3–S9 Figs, S4 Table). Although we detect ongoing interspecific genome-wide gene flow among many species in all three genera [44; S25 Fig], inspection of all significantly associated GWA SNPs at both *ivory* and *optix* regions failed to detect instances where the GWA SNP alleles were shared with congeneric species with mimetic phenotypes (S3–S9 Figs). Twisst [45] and Relate [46] analyses also failed to detect signals of allele sharing (S26–S28 Figs). The only exception is in *Melinaea* where narrow signals of introgression near *optix* are present among multiple species sharing melanic hindwing patterns (S28, S29 Figs). Despite the *ivory* and *optix* regions being repeated targets of selection across multiple species, we do not find evidence of long-term balancing selection maintaining diversity at these regions in *Mechanitis*, *Melinaea*, or *Hypothyris* (S30–S32 Figs). We cannot fully rule out allele sharing because it is possible that our top GWA SNPs do not include causative genetic variants such as indels and structural variants. However, in the ithomiine genera examined, we do not find the relatively long introgressed haplotype blocks that characterize mimetic convergence among closely related *Heliconius* species, despite ongoing interspecific genome-wide gene flow in both groups [22,23]. While most *Heliconius* species have a highly conserved karyotype [47], the frequent karyotypic differences present even among closely related ithomiine species may generate intrinsic postzygotic isolation which limits introgression [44,48,49].

## Repeatable and predictable evolution

The neotropical tiger mimicry ring is exceptionally species-rich, comprising over 100 species from divergent lepidopteran lineages. These butterflies and moths were recognized as a classic example of adaptation by Bates [50], but the genetic basis of their color pattern changes represents a longstanding debate. While Bateson and Punnett [50] argued for large-effect mutations controlling mimicry [51], Fisher [51] predicted that these large-effect loci were likely due to polygenic modifiers scattered across the genome [52]. Nicholson proposed a "two-step" model with a large-effect mutation followed by the fine-tuning of mimicry by polygenic modifiers [53]. Our study clarifies this longstanding dispute: across widely divergent lineages we find two major switch loci, *ivory* and *optix*, with modifiers located next to these switches, rather than scattered across the genome. These modifiers are likely involved in *cis*-regulation of *ivory* and *optix*, and control where on the wing pigmentation changes occur. Phenotypic convergence in all species studied in this mimicry ring is characterized by a simple genetic architecture where a few large-effect genes, primarily *ivory* and *optix*, are reused repeatedly. Even among the most closely related species of the same genus, convergent phenotypic switches occur via independent mutations in the regulatory regions of these genes rather than reuse of standing genetic variation [11] or alleles shared via introgression from other species [13,22]. Similar patterns of gene reuse are found in vertebrate pigmentation, though not necessarily for convergent traits, potentially suggesting that developmental pathways underlying coloration could be more constrained than other traits frequently involved in local adaptation [54,55].

*Ivory* and *optix* are known to control color patterning across Lepidoptera. When these genes control highly similar phenotypes, evolution is surprisingly predictable, with convergence caused by recurrent mutations at very similar regions of these genes. These regions could be mutation hotspots that enable rapid adaptation [56]. The repeatability of evolution also extends to the similar inversion architectures that maintain different allelic combinations in the locally polymorphic butterfly and moth species *Hel. numata* and *C. histrio*. These results suggest that developmental pathways controlling the convergent phenotypes are highly constrained. The different tiger sub-mimicry rings represent locally adaptive fitness peaks. Our results show that not only are the paths to reach these peaks constrained, but also that the steps along these paths tend to be few and large in size, i.e., using large-effect loci. The limited number of paths leading to these fitness peaks may enable diverse taxa to join this species-rich mimicry ring more easily. Once occupying a particular fitness peak, species may then jump via regulatory changes with few pleiotropic effects to alternative peaks representing other locally prevalent color patterns. The outcome of "replaying life's tape" has been a longstanding question in evolutionary biology [3]. Our repeated discovery of convergent adaptation via narrow and repeatable pathways over 120 million years suggests that the running of this tape may be more predictable than expected.

## Methods

### Sample collection and DNA extraction

Samples were collected from across Ecuador, Peru, and Colombia (S17–S23 Figs; S1 Table) between 2002 and 2024. Wings were removed from specimens and kept as vouchers in glassine envelopes. For most samples, the bodies were preserved in NaCl-saturated DMSO solution and stored at −20 °C. Some specimens, including those used for genome assembly, were flash-frozen in liquid nitrogen and stored at −80 °C. DNA extractions were carried out using the Qiagen DNeasy Blood and Tissue Kit, Qiagen MagAttract High Molecular Weight kit, QiaAmp DNA mini kit, or a PureLink digestion and lysis step followed by a magnetic bead DNA extraction [57]. DNA concentration was quantified using Qubit Fluorometer (Invitrogen) and purity assessed using NanoDrop (Thermo Fisher Scientific). For each sample, 350 bp fragment libraries were made using NEBNext Ultra II FS Kit, or using TN5-transposase-mediated tagmentation (adapted from [58]), or following the manufacturer's guidelines with the Illumina DNA PCR-free library prep kit and sequenced (150 bp paired-end) on Illumina NovaSeq 6000 or NovaSeq X machines. In addition, we included Illumina data of *Mechanitis* and *Melinaea* species from van der Heijden and colleagues [43].

**Phenotyping wings**

Wings of specimens were photographed and used to score the color pattern phenotypes. *Chetone histrio* individuals were classed as *Chetone histrio histrio* and *Chetone histrio hydra* with no intermediates (S23 Fig). Yellow band phenotypes in *Hyp. anastasia*, *Mec. messenoides*, *Mel. menophilus*, and *Mel. mothone* were scored as mainly present or absent, with a small number of intermediates (S17–S21 Figs). Hindwing black vs orange in *Mec. messenoides*, *Mel. menophilus, Mel. marsaeus*, and *Hyp. anastasia* were scored as present or absent. The black in the wing tip and base of the forewing in *Mec. messenoides* was scored as present or absent (S22 Fig. The apical spot on the forewing of *Mel. menophilus* was scored as present or absent (S14 Fig). Phenotype scores are shown in S1 Table.

**Quantitative phenotyping of *Mechantis messenoides* black vs orange coloration**

The dorsal side of hindwings of *Mec. messenoides* were photographed within a uniformly lit lightbox. The background was removed from the raw images, and, in cases where a part of the wing was missing, the corresponding wing from the other side was mirrored and used to fill in any gaps. Areas of the three predominant colors (black, yellow, and orange) were then made uniform using CorelDraw. Wings were aligned and the pattern of black coloration was analyzed using Patternize [59]. The extracted black color pattern was analyzed using PCA. PC1 explained 32% of variation in the hindwing black pattern, and the resulting eigenvectors were used as phenotypic values in a GWA analysis (see below).

**Reference genome assembly**

For *Hyp. anastasia* and *Chetone histrio histrio*, DNA was extracted from Ecuadorian flash-frozen females using a standard phenol-chloroform protocol and samples sequenced to ~150× coverage using PromethION R9.4.1 flow cells (Oxford Nanopore Technologies). Genome assembly carried out using NextDenovo 2.5.2 [60] and polished with additional Illumina sequence using NextPolish1.4.1 [61]. The *Hyp. anastasia* assembly was scaffolded with Hi-C data (ERR12071217) generated using the Arima Hi-C+ kit and sequenced on HiSeq X using YaHS [62]. To screen noninsect sequences in the assemblies, we divided the genome into 10 kb windows and performed BLASTN [63] searches against the GenBank complete nt database [64]. Scaffolds with predominantly noninsect matches were removed from the assembly. Additionally, we trimmed scaffolds showing strong similarity to noninsect sequences. Circular mitochondrial assemblies generated from the Illumina reads using GetOrganelle [65] were added to the primary assemblies. Both assemblies showed high contiguity, with scaffold N50 values of 5.8 Mb for *C. histrio histrio* (JBVTKZ010000000) and 37.2 Mb for *Hyp. anastasia* (JBVTLC000000000). Genome completeness was also high, with 97.4% and 98.1% of single-copy BUSCO genes [66] found in *C. histrio histrio* and *Hyp. anastasia* respectively. Additional genome assemblies for *Mel. marsaeus [44, 67]*, *Mec. messenoides [44],* and *Mel. mothone* [44] was used for mapping the respective genome sequence data.

**Whole-genome resequencing and genotype calling**

Illumina adapter sequences were trimmed from the raw FASTQ files using Cutadapt 1.2.1 [68], with further trimming with a minimum window quality score of 20 using Sickle 1.2 [69]. Reads shorter than 15 bp were removed. Trimmed reads were mapped with BWA v0.7.17 [70] using BWA mem and default options against the reference genome of their respective species (Mec. messenoides GCA_959347415.1; *Mel mothone* GCA_965197345.1; *Mel. marsaeus* GCA_918358865.1; *Mel. menophilus* GCA_918358695.1). Mapped reads were sorted using Samtools v1.15 [71], and Picard 2.25.5 (http://broadinstitute.github.io/picard/) was used to add read groups and mark duplicates. The SNP calling was performed using GATK v4.1.3 [72]. The GATKHaplotypeCaller was used to generate GVCF files followed by genotyping using GATK [68] GenotypeGVCFs. VCF files were filtered using Bcftools v1.19 [71] to extract biallelic SNPs with a variant quality score (QUAL) ≥ 10, a genotype quality (GQ) ≥ 10, and a depth of coverage ≥ 5. SNPs with more than 20% of missing data were

removed. We imputed missing data and phased the VCF files for each species using SHAPEIT v4.2 [73] with default parameters.

## BUSCO phylogeny and divergence times

To infer the phylogeny and divergence time between the major groups analyzed in this study, we downloaded additional reference genomes from ENA and built a dated phylogenetic tree of 18 species: *Napeogenes inachia* (GCA_959347435.2)*, Hypothyris anastasia* (this study)*, Ithomia salapia* (GCA_028829205.1)*, Scada zibia* (GCA_964345855.1)*, Methona curvifascia* (GCA_982264085.1)*, Mechanitis messenoides* (GCA_959347415.1)*, Melinaea mothone* (GCA_965197345.1)*, Melinaea menophilus* (GCA_918358695.1)*, Melinaea marsaeus* (GCA_918358865.1)*, Danaus plexippus* (GCA_018135715.1)*, Heliconius pardalinus* (GCA_001486225.1)*, Heliconius melpomene* (Hmel2.5 [74,75])*, Heliconius numata* (GCA_016802625.1)*, Heliconius erato* [76]*, Bicyclus anynana* (GCA_947172395.1)*, Chetone histrio* (this study)*, Biston betularia* (GCA_905404145.2), and *Plutella xylostella* (GCF_932276165.1) using BUSCO genes.

For each reference genome, we ran BUSCO v5.4.3 [66] and extracted the BUSCO genes common to all species. Sequences for each gene were translated into amino acids and aligned using MUSCLE v3.8.31 [77]. The alignment was then reverse-translated to nucleotides using PAL2NAL v14 [78], retaining only genes with fewer than 2% gaps, resulting in a dataset of 257 genes.

We inferred the maximum likelihood (ML) tree based on these concatenated common genes using RAxML v8.2.12 [79,80]. We generated 100 bootstrap alignments using the *-f j* option in RAxML and optimized the model parameters and branch lengths of these bootstrapped trees based on the previously inferred ML tree using the *-f e* option. All trees were rooted using pxrr v1.3.1 [80] with *Plutella xylostella* as the outgroup.

Divergence time estimates were obtained using a penalized-likelihood-based approach implemented in TreePL v1.0 [81]. The node separating Papilionidae from the moths was used as a calibration node, constraining it to range from 100 to 120 Mya [82]. TreePL was run on each bootstrapped tree to obtain age estimate ranges for each node. The priming step was performed on each of the 100 bootstrapped trees, cross-validation was run 10 times, and finally, for the dating step, the best smoothing parameters for each run were chosen based on the lowest $\chi^2$ value and the most common value out of the 10 runs. We used the TreeAnnotator utility from the BEAST package [83] to calculate the 95% highest posterior density for the node ages using a burn-in of 10%.

## Assessing population structure

Population structure within the taxa involved in each genome-wide association analysis was assessed via Principal Component Analysis (PCA) in PLINK v1.9 [84] using an LD pruned SNP dataset. LD pruning was performed in PLINK using a window size of 100 SNPs, a window shift of 10 SNPs and an $r^2$ value of 0.1.

## Genome-wide association mapping

We investigated the genetic basis of phenotypic differences using a genome-wide association (GWA) approach [85]. To identify single-nucleotide polymorphisms (SNPs) associated with each trait, we applied linear univariate mixed models in GEMMA v0.98.5 [86]. SNPs were filtered to retain only those with a minor allele frequency ≥10% and missingness <25%. To account for multiple testing, we applied a Bonferroni correction. Sample relatedness was controlled for by incorporating a pairwise relatedness matrix as a covariate in the model. All other parameters were set to default values.

To annotate the association peak regions, we predicted genes within a 250 kb interval around each peak using AUGUSTUS v3.5.0 [87], trained on the *Hel. melpomene* annotation. We then performed BLASTP [63] searches against the UniProt database [88] to identify and annotate the genes. For SNPs exceeding the significance threshold, we calculated

squared Spearman's rank correlation coefficients ($\rho^2$) to quantify the strength of association between SNP genotypes and phenotypic traits, as both were encoded categorically.

GWA analyses were performed for the presence/absence of a forewing yellow band in *Mel. mothone* ($N=49$), *Mel. menophilus* ($N=64$), *Mec. messenoides* ($N=111$), and *Hyp. anastasia* ($N=61$). For the hindwing, GWA was conducted in *Mel. marsaeus* ($N=40$), *Mec. messenoides* ($N=102$), and *Hyp. anastasia* ($N=61$), comparing solid black versus striped-black phenotypes. For *Mel. menophilus* ($N=67$), we compared three hindwing phenotypes: solid black (encoded as 1), striped-black (0.5), and completely orange (0). For *Mec. messenoides*, hindwing black pattern variation is somewhat more continuous rather than strictly discrete. We also performed an additional GWA using quantitative phenotype values obtained using Patternize [59], producing results consistent with the manual classification (S38 Fig). We also conducted targeted GWA in *Mec. messenoides* to investigate black vs orange pattern variation at both the forewing base ($N=111$) and tip ($N=111$; S10 Fig), as well as in *Mel. menophilus* ($N=67$) to assess the presence/absence of an apical spot on the forewing (S14 Fig).

## GWA peak alignments

For each pair of species, we aligned the regions around the GWA peaks to assess whether these association peaks fall within homologous genomic regions. We limited the alignment to the two genes flanking the GWA peaks. The alignments were performed using Nucmer from the MUMmer package v3.23 [89]. For each genome pair, one genome was divided into nonoverlapping sliding windows of 1,000 bp. These windows were then individually aligned to the alternative genome. Due to the high divergence between genomes, we ran nucmer using the following flags: --mum -c 20 -b 500 -l 10 --maxgap 500.

## QTL mapping in *Heliconius pardalinus*

Crosses and sequencing of hybrids between *Hel. pardalinus butleri* and *Hel. pardalinus sergestus* are described in [90]. Dorsal surfaces of wings from 82 backcross hybrids were photographed in a standardized light box against a white background using a Canon EOS D1000 together with an X-rite ColorChecker Mini to enable color calibration. Yellow and orange forewing patterning, along with orange hindwing patterning were quantified using a standardized patternize workflow as follows. A reference image was selected, and the RGB color signature of a key pattern element was extracted using the sampleRGB() function. Images were aligned to this reference to standardize spatial orientation using the patRegRGB() function, with a color offset (colOffset=0.15) and background removal threshold (removebg=100) to isolate the focal pattern. To quantify variation in color pattern distribution among individuals, we performed a PCA on the aligned pattern rasters using patPCA(). The resulting PCA scores were subsequently used for quantitative trait locus (QTL) mapping as described in [40].

## Butterfly husbandry

Wild *Mec. messenoides messenoides* and *Mec. messenoides deceptus* individuals were caught with hand nets in the Napo province of Ecuador, and used to establish breeding stocks in outdoor insectaries at Ikiam Regional Universidad Amazonica. The adults were fed a solution of sucrose and pollen and provided with *Lantana* and Asteraceae flowers. *Solanum quitoense* was used for oviposition and rearing larvae.

## CRISPR-Cas9 genetic modification

*Ivory* and *optix* were annotated in the reference genome of *Mec. messenoides* (ilMecMess1.1.primary.fa [44]) and *ivory* in *Mel. mothone* (ilMelMoth8.1.primary.fa [44]) based on manual curation of BLAST-hits with the corresponding genes from *Danaus plexippus* and *Hel. erato*. RNA-guides (sgRNA) were designed against the annotated *ivory*- and *optix*-genes using Geneious (www.geneious.com) (S7 Table). Eggs were collected and arranged with nontoxic glue on a microscope

slide. The eggs were injected with a 1:1 mixture of the sgRNA (Sigma-Aldrich) and Cas9-protein (TrueCut Cas9 Protein V2, Invitrogen) at 1 µg/µl within 3–4 hours of laying following established protocols [91].

## In situ hybridization with HCR

Wing tissues were dissected at different developmental timepoints (5th instar caterpillar, day 1−4 after pupation). Caterpillars and pupae were anesthetized on ice before dissection in cold phosphate-buffered saline (PBS). Dissected wing tissue was fixed for 30−40 min with formaldehyde (0.25 ml 37% formaldehyde with 0.75 ml PBS 2 mM ethylene glycol tetraacetic acid), and subsequently dehydrated and stored in methanol at −20 °C following the protocol of [92] until the 'post-fixation' step. Subsequently, the HCR in situ protocol of Molecular Instruments (MI-Protocol-RNAFISH-GenericSolution) was followed. The wings were also stained with DAPI/HOECHST antibody to visualize the nucleus (Sigma-Aldrich; 10236276001), mounted on a slide in 60% glycerol, and imaged with a Leica SP8 confocal microscope.

## Cortex antibody staining

Dissected wing tissues were collected with a fixation (~30−40 min) in 4% paraformaldehyde in PBS 2 mM EGTA. The wings were stored in PT-BSA (PBS 0.1% Triton X-100 with 0.1% sodium azide and Bovine Serum Albumin (0.05 g in 10 ml). The samples were subsequently washed and stained according to a protocol adapted from [93,94]. The primary antibodies against cortex were made for Heliconius (the same antibodies as [94]; rabbit), with secondary antibodies goat anti-rabbit AlexaFluor-555 (ThermoFisher; A-21428). The wings were also stained with Wheat Germ Agglutinin (WGA; plasma membrane; Cambridge BioScience BT29022-1; CF488A Conjugate) at 488 nm and DAPI/HOECHST for the nuclear DNA at 405 nm (Sigma-Aldrich; 10236276001). The tissues were imaged as for the HCR.

## RNA sequencing for annotation of ivory

RNA-seq nanopore long reads were generated for 12 Mel. menophilus wing discs (Day 2 after pupation), comprising three forewing and three hindwing disc samples each from two subspecies: ssp. nov. 1 and hicetas. Nanopore sequences were also generated from wing disc tissues (Day 2 after pupation) for three samples each of Mec. messenoides messenoides, Mec. messenoides deceptus, and Mel. mothone mothone. Wing discs were dissected out and either flash-frozen (Mec. messenoides and Mel. mothone) or stored in RNAlater (Mel. menophilus) at −70 °C until processing. Following extraction, RNA quality was assessed using the Agilent Bioanalyzer. Oxford Nanopore Technologies full-length cDNA sequencing libraries were prepared using the ONT cDNA-PCR Sequencing V14 Barcoding kit (SQK-PCB114.24). Barcoded libraries, with cDNA pooled at equimolar concentrations, were sequenced on R10 flow cells (FLO-PRO114) using an ONT PrometthION sequencer running MinKNOW version 24.02.10. Super-accuracy basecalling was performed with ONT's Dorado software version 7.3.9. The reads from each sample were aligned to their respective reference genomes using minimap2 v2.26 [95] with the command "minimap2 -ax splice -uf -k14". The resulting BAM files were visualized in IGV [96] with the junction track option, focusing on the ivory region. Splicing events supported by at least five reads and present in at least two individuals were retained to define the exons and isoforms of ivory.

## Differential gene expression in Melinaea menophilus

Nanopore reads were mapped to the Mel. menophilus reference genome using Minimap2 v2.26 [95] with parameters optimized for long-read spliced alignment. The resulting BAM files were then used to assemble transcripts for each sample individually with StringTie v3.0.0 [97], using the -L option to account for long reads. These individual transcript assemblies were subsequently merged into a unified transcriptome to generate a consensus annotation. Read quantification, with parameters optimized for long-read data (-L -s 0 -M --fraction -O), was performed using featureCounts v2.0.4 [98] to count reads assigned to genes and isoforms inferred by StringTie. Because automated quantification of long noncoding RNA is often unreliable, the automatically assigned read counts for ivory were manually removed and replaced with curated

counts obtained using IGV. Differential gene expression in *Mel. menophilus* was analyzed by comparing forewing and hindwing datasets for both subspecies (*ssp* versus *hicetas*). Three replicates were included for each condition. Read count data were processed using DESeq2 v1.38.3 [99]. Raw count data were normalized and differential expression was assessed using the Wald test with a Benjamini–Hochberg correction for multiple testing. Genes with an adjusted *p*-value (*padj*) < 0.05 were considered significantly differentially expressed. We then specifically examined whether *ivory* was differentially expressed between the yellow-banded and nonyellow-banded forewings.

## Differential gene expression in *Mechanitis messenoides*

Fore- and hind-wings were dissected at six different timepoints (early 5th instar, late 5th instar, Day 1 after pupation to Day 4 after pupation), for three forewings and hindwings each for both *Mec. messenoides deceptus* and *messenoides* (6 × 3 × 3 × 2 = 72 samples). Each wing was collected separately and flash-frozen in liquid nitrogen. RNA was extracted with a MagMAX Mirvana Total RNA isolation kit, following the manufacturer's protocol (Thermo Fisher Scientific: AM1830). Libraries were 150 bp paired-end Illumina sequenced on two Novaseq X 10B lanes. Illumina reads were trimmed with FastP v0.23.2 [100] and Rcorrector [101], before using STAR without an annotation in 'two pass mode' v2.7.9a [102] to align them to the *Mec. messenoides* reference genome. Based on the STAR-alignments, a de novo transcriptome was assembled using Trinity v2.15.1 [103] with the 'genome_guided_bam' option. Read quantification was performed using Salmon v1.10.2 [104]. Differential expression in the forewing and hindwing datasets was then assessed for both subspecies (*messenoides* versus *deceptus*) using the same approaches as for *Mel. menophilus* (see previous paragraph).

## Measuring genome-wide introgression using *f4* statistics

To investigate potential recent introgression events between sympatric *Melinaea, Mechanitis,* and *Hypothyris* species, we employed the $f_4$-statistics framework implemented in ADMIXTOOLS v2.0.8 [105]. $f_4$-statistics identify deviations in allele frequency correlations across populations or species, which can provide evidence for admixture or introgression. Specifically, we computed $f_4$-statistics between pairs of populations of different species in the same location versus between pairs of populations of the same species in different locations. We designed the test so that a positive $f_4$-statistic indicates excess allele sharing between sympatric species, consistent with gene flow. Statistical significance was assessed using a block-jackknife approach, dividing the genome into nonoverlapping 500 kb blocks.

## Measuring introgression at color genes

We tested for evidence of introgression at wing color pattern genes through inference of local genealogies consistent with introgression, and complemented this approach with visualization of genotypes at top GWA SNPs. These analyses were applied independently to *Mechanitis*, *Melinaea*, and *Hypothyris*, testing whether species with similar phenotypes (i.e., co-mimetic species) showed evidence of gene flow at color loci. $F_d$ and $f_{dM}$ statistics [106,107] did not reveal any evidence of introgression at candidate color loci. We therefore focused on alternative methods that provide more localized or topology-based insights into gene flow in these regions.

To assess introgression at GWA peaks, we employed two complementary approaches to infer local genealogies: (1) marginal trees at individual SNPs using Relate [46], and (2) Neighbor-Joining (NJ) trees built from nonoverlapping 100-SNP sliding windows, using the ape R package [108]. Both methods were applied to a core set of quartets, and some additional quartets were analyzed using the NJ approach (S26–S28 Figs). The two analyses followed a common framework for summarizing topologies and assessing significance.

*Quartet Definition and Taxon Selection*: Analyses were conducted across four different taxon combinations (see S4 Table). Each quartet ideally consisted of two species, each represented by two morphologically distinct subspecies (spA: sspA1/sspA2 and spB: sspB1/sspB2). We specifically focused on introgression between two distant species with similar wing phenotypes, particularly between sspA1 and sspB1. When this configuration was not possible, we modified

 

the arrangement to include one taxon (sspB1) for introgression testing with sspA1 and a third outgroup (sspC1). In both quartet types, we interpreted clustering of sspA1 and sspB1 near the GWA peak as evidence of potential introgression between these species.

*Relate based inference*: Relate was run on 4–10 Mb regions centered on each GWA peak, comparing species involved in the GWA and closely related, phenotypically similar species (S26–S28 Figs). VCFs were imputed and phased with SHAPEIT v4.2 [73], and ancestral alleles were inferred using an outgroup species—*Hyalyris antea* and *Hyalyris lactea* for *Hypothyris*; *Forbestra equicola, F. proceris,* and *F. olivencia* for *Mechanitis*; and *Melinaea ludovica* for *Melinaea*. For SNPs missing in the outgroup, we used the most frequent allele across the genus. Relate was run with an effective population size of $1 \times 10^7$ and a mutation rate of $2.9 \times 10^{-9}$ per site per generation, based on estimates from *Heliconius* studies [109–111]. Because Relate requires a genetic map, we used a uniform recombination rate of 6 cM/Mb, generated with a custom script available at: https://github.com/joanam/scripts/blob/master/createuniformrecmap.r.

*Sliding window NJ tree inference*: As an alternative to Relate, we inferred genealogies by constructing NJtrees from nonoverlapping 100 SNP sliding windows using the ape R package. This window-based approach offered a complementary view of local genealogies by summarizing phylogenetic signal across short genomic regions. For this analysis, we used separate VCFs that retained multiallelic SNPs. NJ tree inference was applied to the same taxon quartets used in the Relate analysis. As this test can be run using a small sample size, additional quartets based on non-GWA species were also analyzed.

*Summarizing tree topologies and assessing significance*: The results from both Relate and the sliding window NJ tree inference were summarized using Twisst [45]. To assess whether a region exhibited a significant excess of introgression-compatible topologies, we performed a permutation test using block-shuffling [112]. This method disrupts potential signals of introgression whilst preserving the local genomic structure, by randomly shuffling 100 kb blocks across the entire region. The null distribution for introgression topology was derived by counting the number of windows with a Twisst weight of at least 0.95 for the introgressed topology (referred to hereafter as intro95). A p-value was calculated by comparing the observed intro95 count in the GWA peak region to the null distribution, based on 50,000 permutations. This approach was applied to both the Relate and sliding window NJ tree analyses to assess introgression significance consistently across both methods.

*Genotypes matrices*: We also created a genotype matrix for the most strongly associated GWA SNPs (S2 Table) to determine whether the same SNPs associated with wing color patterns in one species are also linked to similar phenotypic traits in co-mimetic species. This approach allows us to detect subtle signals of introgression, particularly in cases where the local genealogy analyses might not identify introgression due to the signal being present in a very narrow window.

## Balancing selection

To assess whether color pattern genes are targets of long-term balancing selection, we examined patterns of polymorphism and allele frequency across *Mechanitis*, *Melinaea*, and *Hypothyris* species. To test for balancing selection at the color genes of interest across species, we used three approaches: multispecies nucleotide diversity, detection of *trans*-species polymorphisms, and the MuteBaSS method [113]. For *Melinaea*, analyses were conducted at *ivory*, *optix*, and *antp*, whereas in *Mechanitis* and *Hypothyris*, analyses focused on *ivory* and *optix*.

*Multispecies nucleotide diversity*: To quantify polymorphism at color genes and surrounding regions, we calculated multispecies nucleotide diversity by pooling data across species and subspecies. We used Pixy v1.2.5 [114] to compute nucleotide diversity in nonoverlapping 50 kb sliding windows. Analyses were performed on repeat-masked VCF files including invariant sites as well as bi- and multiallelic SNPs. Repetitive elements in the Lepidoptera genomes were identified using RepeatModeler v2.0.4 [115], and subsequently masked with RepeatMasker v4.1.2 [116] using the generated repeat library. We calculated nucleotide diversity under two conditions:

1. Analysis including all species.

2. Pairwise comparisons between individual species/subspecies to identify cases where balancing selection signals are limited to a subset of taxa.

Detailed species groupings for *Melinaea*, *Mechanitis*, and *Hypothyris* are provided in S6 Table.

*Trans-species polymorphisms*: To test for *trans*-species polymorphisms across multiple species, we analyzed genomic variation shared across multiple species using a custom python pipeline based on *pysam* v0.21.0 (https://github.com/pysam-developers/pysam). We used the same repeat-masked VCF files as described in the 'multispecies nucleotide diversity' section.

A site was classified as transpolymorphic if it met the following criteria: i) at least four species were polymorphic (when more than three species were present); ii) all species were polymorphic (if three or fewer species were present); iii) each allele was represented by at least six total copies across species.

*MuteBaSS*: To further test for local signals of balancing selection, we used MuteBaSS v1.0 [113], which detects balancing selection without relying on *trans*-species polymorphic sites. We used MuteBaSS to compute the multispecies NCD, NCDsub, and NCDopt statistics, as well as the *trans*-HKA test. To capture the potentially narrow signals of balancing selection, all statistics were calculated in 1 kb windows with a 500 bp sliding step. Ancestral states were inferred as described in the *Relate based inference* section. For *Melinaea* and *Mechanitis*, we assumed SNPs followed the phylogeny proposed by Van der Heijden and colleagues [44]. For *Hypothyris*, we used the phylogeny from Chazot and colleagues [117].

## Transcription factor binding site analysis

In all cases excluding the inversion in *Chetone histrio*, the color pattern associated regions identified by GWA fall outside of the genes of interest. This may suggest that the associated regions control color pattern by influencing gene regulation, potentially by affecting transcription factor binding. As the peaks of association with the yellow band phenotype are narrow and contain only a few fixed SNPs, it may be possible to associate these SNPs with a specific transcription factor binding site, by identifying binding motifs which are enriched consequences of one color pattern form compared to the other. To test this, we focused on our best sampled species, *Mec. messenoides*. Multiple analyses were performed to uncover transcription factor (TF) binding sites within and around the *ivory* GWA peak region, defined here as the ~1.5 kb region between the weakly-associated SNPs which flank the associated region identified through GWA (S2 Fig). Only individuals homozygous for the fixed SNPs identified in the GWA analysis were selected. A fasta file for the region of interest was generated for each individual using the consensus commands in either samtools or bcftools. Homer [118] was used to identify de novo DNA motifs which are enriched in one set of sequences compared to a control group, using the findMotifs.pl script and the insect known motif collection. The two forms (yellow-banded versus nonyellow-banded) were alternately used as both control and target sequences.

To locate and identify known TF binding sites within the peak region, the FIMO algorithm in MEME Suite [119] was run on all sequences, including both forms, for each species. A background file was generated based on the entire chromosome containing the peak region (CHROM6), using the fasta-get-markov command. The expression of detected motifs was checked by examining expression levels of the corresponding transcription factors within the *Mec. messenoides* RNAseq data.

## Ethics statement

We thank SERFOR, the Peruvian Ministry of Agriculture and the Área de Conservación Regional Cordillera Escalera (0289-2014-MINAGRI-DGFFS/DGEFFS, 020-014/GRSM/PEHCBM/DMA/ACR-CE and 040–2015/GRSM/PEHCBM/DMA/ACR-CE) for collecting permits; the Ministerio del Ambiente and Museo Ecuatoriano de Ciencias Naturales in Ecuador

(MAATE-DBI-CM-2021-176 and MAATE-DBI-CM-2023-0286) for collecting and butterfly rearing permits. Field collections in Colombia were conducted under permit no. 530 issued by the Autoridad Nacional de Licencias Ambientales (ANLA) of Colombia.

## Supporting information

**S1 Fig. Key to butterfly images used in** [Fig 1a](). 1). *Melinaea menophilus zaneka* 2). *Hypothyris moebiusi moebiusi* 3). *Hypothyris mamercus mamercus* 4). *Heliconius numata euphone* 5). *Eresia pelonia f. callonia* 6). *Melinaea isocomma isocomma* 7). *Consul fabius bogatanus* 8). *Mechanitis mazeus fallax* 9). *Forbestra equicola* 10). *Hypothyris mansuetus amica* 11). *Themone paid trivittata* 12). *Hypothyris cantobrica zamorita* 13). *Hypothyris semifulva ssp.* 14). *Melinaea mnasias abitagua* 15). *Mechanitis messenoides messenoides* 16). *Hypothyris euclea pyrippe* 17). *Melinaea marseus messenina* 18). *Chetone histrio histrio* 19). *Heliconius numata bicoloratus* 20). *Hypothyris semifulva semifulva* 21). *Eueides lampeto* 22). *Eresia pelonia f. Ithomiola* 23). *Napeogenes rhezia acaea* 24). *Hypothyris anastasia aureata* 25). *Hypothyris mansuetus meterus* 26). *Hyposcada anchiala mendax* 27). *Melinaea mothone* 28). *Mechanitis messenoides deceptus* 29). *Hypothyris anastasia bicolora* 30). *Melinaea isocomma simulator* 31). *Chetone histrio hydra* 32). *Hypothyris euclea callanga* 33). *Hypothyris fluonia seminigra* 34). *Forbestra olivencia olivencia* 35). *Melinaea menophilus orestes* 36). *Melinaea mnasias romualdo* 37). *Tithorea harmonia brunnea* 38). *Napeogenes zurippa deucalion* 39). *Heliconius pardalinus butleri* 40). *Melinaea marseus phasiana* 41). *Hypothyris anastasia anastasina* 42). *Melinaea menophilus hicetas* 43). *Mechanitis mazeus mazeus* 44). *Chetone histrio histrio* 45). *Hypothyris anastasia acreana* 46). *Melinaea satevis lamasi* 47). *Athyrtis mechanitis salvini*, and 48). *Melinaea marseus clara*. Some butterfly images courtesy of https://www.butterfliesofamerica.com/ (Andrew Warren); http://www.sangay.eu/esdex.php/ (Jean-Claude Petit). Maps were generated using openly available base maps from Stadia Maps (stadiamaps.com), with the Stamen Terrain style (stamen.com).
(TIF)

**S2 Fig. Genome-wide associations for forewing yellow band and hindwing black vs orange wing phenotypes.** SNPs above the Bonferroni-corrected significance threshold (horizontal dashed line) in the main peak of association are highlighted in orange. The wing images on the right denote the phenotype compared in each analysis. Zoomed-in plots of the peaks are shown in [Fig 2](). Additional associated SNPs in some of the species are likely a result of population structure correlated with the phenotypes ([S17]()–[S20 Figs]()). The underlying data can be found at https://zenodo.org/records/19135682.
(TIFF)

**S3 Fig. Genotypic variation at the *Mechanitis messenoides ivory* GWA peak in *Mechanitis* species.** This genotype matrix shows the genotypic states at the top SNPs from the GWA at the *ivory* locus, with individuals grouped by phylogenetic relationships (left dendrogram). Each row corresponds to an individual and each column to a SNP. The top panels (down to *messenoides messenoides*) include the focal species used in the GWAS. The remaining taxa are shown to illustrate the lack of association between genotype and phenotype at these SNPs across a broader phylogenetic context. Phenotype group (wing color patterns) is represented by colored boxes to the left of the genotype matrix. The numbers displayed along the top of the figure are the genomic positions of the SNPs. The GWA -log(*p*-value) values are displayed beneath the genomic positions, with colors ranging from yellow (low significance) to dark purple (high significance). The underlying data can be found at https://zenodo.org/records/19135682.
(TIFF)

**S4 Fig. Genotypic variation at the *Melinaea mothone ivory* GWA peak in *Melinaea* species.** This genotype matrix shows the genotypic states at the top SNPs from the GWA at the *ivory* locus, with individuals grouped by phylogenetic relationships (left dendrogram). Each row corresponds to an individual and each column to a SNP. The top panels (down

to *mothone messenina*) include the focal species used in the GWAS. The remaining taxa are shown to illustrate the lack of association between genotype and phenotype at these SNPs across a broader phylogenetic context. Phenotype group (wing color patterns) is represented by colored boxes to the left of the genotype matrix. The numbers displayed along the top of the figure are the genomic positions of the SNPs. The GWA -log(*p*-value) values are displayed beneath the genomic positions, with colors ranging from yellow (low significance) to dark purple (high significance). The underlying data can be found at https://zenodo.org/records/19135682.
(TIFF)

**S5 Fig. Genotypic variation at the *Melinaea menophilus ivory* GWA peak in *Melinaea* species.** This genotype matrix shows the genotypic states at the top SNPs from the GWA at the *ivory* locus, with individuals grouped by phylogenetic relationships (left dendrogram). Each row corresponds to an individual and each column to a SNP. The top panels (down to *menophilus zaneka*) include the focal species used in the GWAS. The remaining taxa are shown to illustrate the lack of association between genotype and phenotype at these SNPs across a broader phylogenetic context. Phenotype group (wing color patterns) is represented by colored boxes to the left of the genotype matrix. The numbers displayed along the top of the figure are the genomic positions of the SNPs. The GWA -log(*p*-value) values are displayed beneath the genomic positions, with colors ranging from yellow (low significance) to dark purple (high significance). The underlying data can be found at https://zenodo.org/records/19135682.
(TIFF)

**S6 Fig. Genotypic variation at the *Hypothyris anastasia ivory* GWA peak in *Hypothyris* species.** This genotype matrix shows the genotypic states at the top SNPs from the GWA at the *ivory* locus, with individuals grouped by phylogenetic relationships (left dendrogram). Each row corresponds to an individual and each column to a SNP. The top panels (down to *anastasia bicolora*) include the focal species used in the GWAS. The remaining taxa are shown to illustrate the lack of association between genotype and phenotype at these SNPs across a broader phylogenetic context. Phenotype group (wing color patterns) is represented by colored boxes to the left of the genotype matrix. The numbers displayed along the top of the figure are the genomic positions of the SNPs. The GWA -log(*p*-value) values are displayed beneath the genomic positions, with colors ranging from yellow (low significance) to dark purple (high significance). The underlying data can be found at https://zenodo.org/records/19135682.
(TIFF)

**S7 Fig. Genotypic variation at the *Melinaea marsaeus optix* GWA peak in *Melinaea* species.** This genotype matrix shows the genotypic states at the top SNPs from the GWA at the *optix* locus, with individuals grouped by phylogenetic relationships (left dendrogram). Each row corresponds to an individual and each column to a SNP. The top panels (down to *marsaeus phasiana*) include the focal species used in the GWAS. The remaining taxa are shown to illustrate the association between genotype and phenotype at these SNPs across a broader phylogenetic context. Phenotype group (wing color patterns) is represented by colored boxes to the left of the genotype matrix. The numbers displayed along the top of the figure are the genomic positions of the SNPs. The GWA -log(*p*-value) values are displayed beneath the genomic positions, with colors ranging from yellow (low significance) to dark purple (high significance). The underlying data can be found at https://zenodo.org/records/19135682.
(TIFF)

**S8 Fig. Genotypic variation at the *Melinaea menophilus optix* GWA peak in *Melinaea* species.** This genotype matrix shows the genotypic states at the top SNPs from the GWA at the *optix* locus, with individuals grouped by phylogenetic relationships (left dendrogram). Each row corresponds to an individual and each column to a SNP. The top panels (down to *menophilus zaneka*) include the focal species used in the GWAS. The remaining taxa are shown to illustrate the lack of association between genotype and phenotype at these SNPs across a broader phylogenetic context. Phenotype group

 

(wing color patterns) is represented by colored boxes to the left of the genotype matrix. The numbers displayed along the top of the figure are the genomic positions of the SNPs. The GWA -log($p$-value) values are displayed beneath the genomic positions, with colors ranging from yellow (low significance) to dark purple (high significance). The underlying data can be found at https://zenodo.org/records/19135682.
(TIFF)

**S9 Fig. Genotypic variation at the *Hypothyris anastasia optix* GWA peak in *Hypothyris* species.** This genotype matrix shows the genotypic states at the top SNPs from the GWA at the *optix* locus, with individuals grouped by phylogenetic relationships (left dendrogram). Each row corresponds to an individual and each column to a SNP. The top panels (down to *anastasia acreana*) include the focal species used in the GWAS. The remaining taxa are shown to illustrate the lack of association between genotype and phenotype at these SNPs across a broader phylogenetic context. Phenotype group (wing color patterns) is represented by colored boxes to the left of the genotype matrix. The numbers displayed along the top of the figure are the genomic positions of the SNPs. The GWA -log($p$-value) values are displayed beneath the genomic positions, with colors ranging from yellow (low significance) to dark purple (high significance). The underlying data can be found at https://zenodo.org/records/19135682.
(TIFF)

**S10 Fig. Genome-wide associations for different black vs orange wing phenotypes in *Mechanitis messenoides*.** Genome-wide associations for black and orange wing patterns are shown for three different wing regions: forewing base, forewing tip, and hindwing melanization (wing images show the phenotype compared in each analysis). SNPs above the Bonferroni-corrected significance threshold (horizontal dashed line) in the main peak of association are highlighted in orange, which in all cases lies near *optix*. A zoomed-in plot of the peak is shown in S4A Fig. The underlying data can be found at https://zenodo.org/records/19135682.
(TIFF)

**S11 Fig. Genotypic variation at the *Mechanitis messenoides optix* forewing base GWA peak in *Mechanitis* species.** This genotype matrix shows the genotypic states at the top SNPs from the GWA at the *optix* locus, with individuals grouped by phylogenetic relationships (left dendrogram). Each row corresponds to an individual and each column to a SNP. The top three panels include the focal species used in the GWAS. The remaining taxa are shown to illustrate the lack of association between genotype and phenotype at these SNPs across a broader phylogenetic context. Phenotype group (wing color patterns) is represented by colored boxes to the left of the genotype matrix. The numbers displayed along the top of the figure are the genomic positions of the SNPs. The GWA -log($p$-value) values are displayed beneath the genomic positions, with colors ranging from yellow (low significance) to dark purple (high significance). The underlying data can be found at https://zenodo.org/records/19135682.
(TIFF)

**S12 Fig. Genotypic variation at the *Mechanitis messenoides optix* forewing tip GWA peak in *Mechanitis* species.** This genotype matrix shows the genotypic states at the top SNPs from the GWA at the *optix* locus, with individuals grouped by phylogenetic relationships (left dendrogram). Each row corresponds to an individual and each column to a SNP. The top three panels include the focal species used in the GWAS. The remaining taxa are shown to illustrate the lack of association between genotype and phenotype at these SNPs across a broader phylogenetic context. Phenotype group (wing color patterns) is represented by colored boxes to the left of the genotype matrix. The numbers displayed along the top of the figure are the genomic positions of the SNPs. The GWA -log($p$-value) values are displayed beneath the genomic positions, with colors ranging from yellow (low significance) to dark purple (high significance). The underlying data can be found at https://zenodo.org/records/19135682.
(TIFF)

**S13 Fig. Genotypic variation at the *Mechanitis messenoides optix* hindwing GWA peak in *Mechanitis* species.** This genotype matrix shows the genotypic states at the top SNPs from the GWA at the *optix* locus, with individuals grouped by phylogenetic relationships (left dendrogram). Each row corresponds to an individual and each column to a SNP. The top three panels include the focal species used in the GWAS. The remaining taxa are shown to illustrate the lack of association between genotype and phenotype at these SNPs across a broader phylogenetic context. Phenotype group (wing color patterns) is represented by colored boxes to the left of the genotype matrix. The numbers displayed along the top of the figure are the genomic positions of the SNPs. The GWA -log(*p*-value) values are displayed beneath the genomic positions, with colors ranging from yellow (low significance) to dark purple (high significance). The underlying data can be found at https://zenodo.org/records/19135682.
(TIFF)

**S14 Fig. Genome-wide association for the forewing apical spot in *Melinaea menophilus*.** **(A)** Manhattan plot for the whole-genome. The orange dashed line represents the threshold of significance. SNPs in the main peak of association are highlighted in orange. **(B)** Zoom on the main peak of association (SUPER_4:16233326–16458737) with locations of annotated genes. SNPs above the Bonferroni-corrected significance threshold (dashed orange line) are colored according to the squared Spearman's rank correlation coefficient ($\rho^2$), which indicates the strength of association between genotype and phenotype. Red arrow: position of Topologically Associated Domain boundary element *Antp-Ubx_BE*[34]. **(C)** The wing phenotypes compared in the analysis. **(D)** Principal component analysis of 412,584 LD pruned biallelic SNPs showing the population structure among the sampled individuals. Points are colored by apex spot phenotype. The underlying data can be found at https://zenodo.org/records/19135682.
(TIFF)

**S15 Fig. Genotypic variation at the *Melinaea menophilus antennapedia* GWA peak in *Melinaea* species.** This genotype matrix shows the genotypic states at the top SNPs from the GWA at the *antennapedia* locus, with individuals grouped by phylogenetic relationships (left dendrogram). Each row corresponds to an individual and each column to a SNP. The top panels (down to *menophilus zaneka*) include the focal species used in the GWAS. The remaining taxa are shown to illustrate the lack of association between genotype and phenotype at these SNPs across a broader phylogenetic context. Phenotype group (wing color patterns) is represented by colored boxes to the left of the genotype matrix. The numbers displayed along the top of the figure are the genomic positions of the SNPs. The GWA -log(*p*-value) values are displayed beneath the genomic positions, with colors ranging from yellow (low significance) to dark purple (high significance). The underlying data can be found at https://zenodo.org/records/19135682.
(TIFF)

**S16 Fig. QTL mapping intervals for forewing and hindwing pattern variation in *Heliconius pardalinus*.** **(A)** Mapping intervals for yellow and orange forewing and hindwing pattern variation. Heatmap images to the left indicate the phenotypes exhibited by the individuals at the extremes of each PC axis; red = presence of the color, blue = absence of the color. Note that the forewing has only yellow/white, orange/red, and black/brown colors, and the hindwing has only orange/red and black/brown. Since the black/brown pattern variation is a complement of the other colors, we do not show the results of the black/brown variation. **(B)** Details of the QTL mapping intervals. The underlying data can be found at https://zenodo.org/records/19135682.
(TIFF)

**S17 Fig. *Hypothyris anastasia*: taxon distribution, wing phenotypes and genetic PCA.** **(A)** Geographical distribution of *Hypothyris anastasia* subspecies, with specimen collection locations indicated. Map produced using Natural Earth base layer (https://www.naturalearthdata.com/downloads/50m-cultural-vectors/50m-admin-0-countries-2/) **(B)** Phenotypes used in the GWA. **(C to F)**: Principal component analysis (PCA) showing the genetic distance between the sampled *Hypothyris*

*anastasia* individuals using a dataset consisting of 400,452 LD pruned biallelic SNPS. The scatter plots correspond to the first two principal components (PCs). Points are colored by subspecies **(C)**, geography **(D)**, yellow band phenotype **(E)**, and hindwing black phenotype **(D)**. The wing images highlight the color-coded phenotypes. The underlying data can be found at https://zenodo.org/records/19135682.
(TIFF)

**S18 Fig. *Melinaea menophilus*: taxon distribution, wing phenotypes, and genetic PCA. (A)** Geographical distribution of *Melinaea menophilus* subspecies, with specimen collection locations indicated. Map produced using Natural Earth base layer (https://www.naturalearthdata.com/downloads/50m-cultural-vectors/50m-admin-0-countries-2/) **(B)** Phenotypes used in the GWA. **(C to F)**: Principal Principal component analysis (PCA) showing the genetic distance between the sampled *Melinaea menophilus* individuals using a dataset consisting of 412,584 LD pruned biallelic SNPS. The scatter plots correspond to the first two principal components (PCs). Points are colored by subspecies **(C)**, geography **(D)**, yellow band phenotype **(E),** and hindwing black phenotype **(F)**. The wing images highlight the color-coded phenotypes. The underlying data can be found at https://zenodo.org/records/19135682.
(TIFF)

**S19 Fig. *Melinaea marseus*: taxon distribution, wing phenotypes, and genetic PCA. (A)** Geographical distribution of *Melinaea marsaeus* subspecies, with specimen collection locations indicated. Map produced using Natural Earth base layer (https://www.naturalearthdata.com/downloads/50m-cultural-vectors/50m-admin-0-countries-2/) **(B)** Phenotypes used in the GWA. **(C–F)**: Principal component analysis (PCA) showing the genetic distance between the sampled *Melianea marsaeus* individuals using a dataset consisting of 205,438 LD pruned biallelic SNPs (C–E) or 1,686 no LD pruned SNPs around the *Optix* region (F). The scatter plots correspond to the first two principal components (PCs). Points are colored by subspecies (C), geography (D), hindwing black phenotype (E, F). The wing images highlight the color-coded phenotypes. The underlying data can be found at https://zenodo.org/records/19135682.
(TIFF)

**S20 Fig. *Melinaea mothone*: taxon distribution, wing phenotypes, and genetic PCA. (A)** Geographical distribution of *Melinaea mpothone* subspecies, with specimen collection locations indicated. Map produced using Natural Earth base layer (https://www.naturalearthdata.com/downloads/50m-cultural-vectors/50m-admin-0-countries-2/ **(B)** Phenotypes used in the GWA. **(C–E)**: Principal component analysis (PCA) showing the genetic distance between the sampled *Melinaea mothone* individuals using a dataset consisting of 13,652 LD pruned biallelic SNPs. The scatter plots correspond to the first two principal components (PCs). Points are colored by subspecies (C), geography (D), and yellow band phenotype (E). The wing images highlight the color-coded phenotypes. The underlying data can be found at https://zenodo.org/records/19135682.
(TIFF)

**S21 Fig. *Mechanitis messenoides* (forewing yellow band and hindwing black vs orange): taxon distribution, wing phenotypes, and genetic PCA. (A)** Geographical distribution of *Mechanitis messenoides* subspecies, with specimen collection locations indicated. Map produced using Natural Earth base layer (https://www.naturalearthdata.com/downloads/50m-cultural-vectors/50m-admin-0-countries-2/ (B) Phenotypes used in the GWA. **(C–F)**: Principal component analysis (PCA) showing the genetic distance between the sampled *Mechanitis messenoides* individuals using a dataset consisting of 86,312 LD pruned biallelic SNPS. The scatter plots correspond to the first two principal components (PCs). Points are colored by subspecies (C), geography (D), yellow band phenotype (E), and hindwing black phenotype (F). The wing images highlight the color-coded phenotypes. The underlying data can be found at https://zenodo.org/records/19135682.
(TIFF)

**S22 Fig. *Mechanitis messenoides* (forewing base and tip black vs orange): taxon distribution, wing phenotypes, and genetic PCA. (A)** Geographical distribution of *Mechanitis messenoides* subspecies, with specimen collection locations indicated. Map produced using Natural Earth base layer (https://www.naturalearthdata.com/downloads/50m-cultural-vectors/50m-admin-0-countries-2/ **(B)** Phenotypes used in the GWA. **(C–D):** Principal component analysis (PCA) showing the genetic distance between the sampled *Mechanitis messenoides* individuals using a dataset consisting of 86,312 LD pruned biallelic SNPS. The scatter plots correspond to the first two principal components (PCs). Points are colored by forewing base phenotype (C) and forewing tip phenotype (D). The wing images highlight the color-coded phenotypes. The underlying data can be found at https://zenodo.org/records/19135682.
(TIFF)

**S23 Fig. *Chetone histrio*: taxon distribution, wing phenotypes, and genetic PCA. (A)** Geographical distribution of *Chetone histrio* subspecies, with specimen collection locations indicated. Map produced using Natural Earth base layer (https://www.naturalearthdata.com/downloads/50m-cultural-vectors/50m-admin-0-countries-2/ **(B)** Phenotypes used in the GWA. Principal Component Analysis (PCA) of 1,159,948 LD pruned biallelic SNPs in *Chetone histrio* with points colored by **(C)** geography and **(D)** subspecies. The underlying data can be found at https://zenodo.org/records/19135682.
(TIFF)

**S24 Fig. Inversion breakpoint analysis in *Chetone histrio*.** Examination of Illumina read-pair orientation and mapped insert sizes using IGV allows localization of the inversion breakpoints in *Chetone histrio.* The IGV screenshots are for individual NR15−519, which is heterozygous for the inversion. a) Comparison of the locations of the *Chetone histrio* and *Heliconius numata* P1 inversion based on the locations of genes labeled 0−34. b) The left breakpoint in *Chetone histrio* is located between genes 0 and 1 at ctg001860:9,853,770−9,855,878. A region of poor mapping prevents more accurate inference of the breakpoint. c) The right breakpoint in *Chetone histrio* is located between genes 33 and 34 at ctg001860:10,873,356. Gene 0–34 refer to: 0: *beta-fructofuranosidase*; 1: *glutaminyl-peptidecyclotransferase*; 2: *HMEL000021*; 3: *enoyl-CoA hydratase*; 4: *Cancer-related nucleoside-triphosphatase*; 5: *Sur-8/LRR*; 6: *HMEL032678*; 7: *HMEL002023g1*; 8: *HMEL002023g2*; 9: *HMEL002024*; 10: *cortex*; 11: *parn*; 12: *HMEL000027*; 13: *ARP-like*; 14: *ATP synthase subunit f*; 15: *proteasome 26S non ATPasesubunit 4*; 16: *zinc phosphodiesterase*; 17: *serine/threonine-proteinkinase*; 18: *WD repeat-containing protein 19*; 19: *HMEL013472*; 20: *WAS protein family homologue 1*; 21: *Domeless*; 22: *HMEL032681*; 23: *HMEL032683*; 24: *mitogen-activated protein kinase*; 25: *DNA excision repair protein ERCC-6*; 26: *penguin*; 27: t*hymidylate kinase*; 28: *caspase-activated DNase*; 29: *ribosome biogenesis regulatory protein*; 30: *INO80 complex subunit C*; 31: *uncharacterized WD repeat-containing protein C2E1P5.05*; 32: *Sr protein*; 33: *HMEL000048*; 34: *HMEL000049*.
(TIFF)

**S25 Fig. $f_4$ statistics testing for allele sharing between sympatric species of *Hypothyris*, *Mechanitis*, and *Melinaea.*** Comparisons are shown across different geographic regions (Colombia-Ecuador, Colombia-Peru, and Ecuador-Peru). Each point represents an $f_4$ value, with standard errors. Red points are significantly different from zero ($p < 0.05$). Significantly positive $f_4$ value suggests excess allele sharing between the tested species, consistent with interspecific gene flow. Details of the taxa tested are shown in S5 Table. The underlying data can be found at https://zenodo.org/records/19135682.
(TIFF)

**S26 Fig. Testing for introgression across pairs of *Mechanitis* species at A) *ivory* and B) *optix*.** Comparisons are shown between species with different forewing yellow band (top) and hindwing black vs orange (bottom) phenotypes. Each cell in the matrix represents a comparison between a pair of species, with rows and columns labeled by species and wing phenotype. No evidence of introgression (pink) is found in any comparison. The Twisst and Relate p-values,

based on a block permutation test, are displayed in each cell. "NA" indicates intraspecific or invalid comparisons (gray). *P*-values of 1.00000 indicate that no introgression-compatible topologies were observed within the GWAS peak region, making it impossible to compute a *p*-value. For Relate, "low" indicates that the analyses included fewer than 20 samples and could not be run. Species and wing phenotypes are depicted along the matrix's margins with corresponding wing photographs. Details of the taxa tested are shown in S4 Table. The underlying data can be found at https://zenodo.org/records/19135682.
(TIF)

**S27 Fig. Testing for introgression across pairs of *Hypothyris* species at A) *ivory* and B) *optix*.** Comparisons are shown between species with different forewing yellow band (top) and hindwing black vs orange (bottom) phenotypes. Each cell in the matrix represents a comparison between a pair of species, with rows and columns labeled by species and wing phenotype. No evidence of introgression (pink) is found in any comparison. The Twisst and Relate p-values, based on a block permutation test, are displayed in each cell. "NA" indicates intraspecific or invalid comparisons (gray). *P*-values of 1.00000 indicate that no introgression-compatible topologies were observed within the GWAS peak region, making it impossible to compute a *p*-value. For Relate, "low" indicates that the analyses included fewer than 20 samples and could not be run. Species and wing phenotypes are depicted along the matrix's margins with corresponding wing photographs. Details of the taxa tested are shown in S4 Table. The underlying data can be found at https://zenodo.org/records/19135682.
(TIF)

**S28 Fig. Testing for introgression across pairs of *Melinaea* species at A) *ivory*, B) *optix*, and C) *antennapedia*.** Comparisons are shown between species with different forewing yellow band (top), hindwing black vs orange (middle), and forewing apex spot (bottom) phenotypes. Each cell in the matrix represents a comparison between a pair of species, with rows and columns labeled by species and wing phenotype. The matrix is color-coded to indicate the type of evidence detected: no evidence of introgression (pink), evidence from Relate only (orange), evidence from Twisst only (ochre), or evidence from both Relate and Twisst (blue). The Twisst and Relate *p*-values, based on a block permutation test, are displayed in each cell. "NA" indicates intraspecific or invalid comparisons (gray). *P*-values of 1.00000 indicate that no introgression-compatible topologies were observed within the GWAS peak region, making it impossible to compute a *p*-value. For Relate, "low" indicates that the analyses included fewer than 20 samples and could not be run. Species and wing phenotypes are depicted along the matrix's margins with corresponding butterfly illustrations. Abbreviations for species names are as follows: *meno*.: *menophius*; *mars*.: *marsaeus*; *isoc*.: *isocomma*; *moth*.: *mothone*. Details of the taxa tested are shown in S4 Table. The underlying data can be found at https://zenodo.org/records/19135682.
(TIF)

**S29 Fig. Genomic signals of introgression at the *optix* locus among four pairs of *Melinaea* species. (A)** *Melinaea marsaeus* GWA Manhattan plot near *optix*. The black rectangle highlights the strongest association peak, which is shown zoomed in on the right panel. **(B)** Twisst weight bar plots displaying local phylogenetic topology support across the highlighted region in a. Each row corresponds to a different species pair, with the y-axis indicating the Twisst weight, which quantifies how congruent the local genealogy is with one of the three topologies shown in **(C)** The left bar plots provide a broader genomic context and include a smoothing function to reduce noise, while the right bar plots shows a zoom in of the region without smoothing. Higher red values indicate greater support for an introgressed topology, while the two shades of gray correspond to the two alternative topologies expected for unrooted trees based on four taxa. (C) Most likely marginal coalescent trees for each species pair tested, inferred using *RELATE*. The underlying data can be found at https://zenodo.org/records/19135682.
(TIFF)

**S30 Fig. Multispecies balancing selection tests in *Melinaea* at *antennapedia*, *ivory* and *optix* (left to right).** These analyses included *Melinaea isocomma*, *Melinaea mothone*, *Melinaea marsaeus*, *Melinaea menophilus*, *Melinaea satevis*, and *Melinaea tarapotensis*. The red-shaded regions indicate the location of the GWA peaks in the genes *antennapedia*, *ivory* and *optix*. **(A)** HKAtrans statistics computed in 1 kb windows with a 500 bp sliding step. *Melinaea ludovica* was used to determine the ancestral state. Positive HKAtrans values indicate an excess of shared polymorphisms relative to divergence, which is consistent with ancient *trans*-species balancing selection. **(B)** NCD2$_{trans}$ statistics computed in 1 kb windows with a 500 bp sliding step. This statistic measures allele frequency deviations from neutral expectations across multiple species. Values closer to 0 suggest ancient *trans*-species balancing selection. **(C)** NCD2$_{trans\text{-}opt}$ statistics computed in 1 kb windows with a 500 bp sliding step. This statistic is a variant of NCD2$_{trans}$ that optimizes the target frequency at neutrality. Values closer to 0 suggest ancient *trans*-species balancing selection. **(D)** NCD2$_{trans\text{-}sub}$ statistics computed in 1 kb windows with a 500 bp sliding step. A variant of NCD2$_{trans}$ that treats substitutions and polymorphisms separately. Values closer to 0 suggest ancient *trans*-species balancing selection. **(E)** Multispecies nucleotide diversity (π) computed in 10 kb windows. Larger values are indicative of ancient *trans*-species balancing selection. **(F)** Number of transpolymorphic sites computed in 10 kb windows. Larger values are indicative of ancient *trans*-species balancing selection. The underlying data can be found at https://zenodo.org/records/19135682.
(TIFF)

**S31 Fig. Multispecies balancing selection tests in *Hypothyris* at *ivory* and *optix* (left to right).** These analyses included *Hypothyris anastasia*, *Hypothyris euclea*, *Hypothyris ninonia*, and *Hypothyris semifulva*. The red-shaded regions indicate the location of the GWA peaks in the genes *ivory* and *optix*. **(A)** HKAtrans statistics computed in 1 kb windows with a 500 bp sliding step. *Hyalyris antea* was used to determine the ancestral state. Positive HKAtrans values indicate an excess of shared polymorphisms relative to divergence, which is consistent with ancient *trans*-species balancing selection. **(B)** NCD2$_{trans}$ statistics computed in 1 kb windows with a 500 bp sliding step. This statistic measures allele frequency deviations from neutral expectations across multiple species. Values closer to 0 suggest ancient *trans*-species balancing selection. **(C)** NCD2$_{trans\text{-}opt}$ statistics computed in 1 kb windows with a 500 bp sliding step. This statistic is a variant of NCD2$_{trans}$ that optimizes the target frequency at neutrality. Values closer to 0 suggest ancient *trans*-species balancing selection. **(D)** NCD2$_{trans\text{-}sub}$ statistics computed in 1 kb windows with a 500 bp sliding step. A variant of NCD2$_{trans}$ that treats substitutions and polymorphisms separately. Values closer to 0 suggest ancient *trans*-species balancing selection. **(E)** Multispecies nucleotide diversity (π) computed in 10 kb windows. Larger values are indicative of ancient *trans*-species balancing selection. **(F)** Number of transpolymorphic sites computed in 10 kb windows. Larger values are indicative of ancient *trans*-species balancing selection. The underlying data can be found at https://zenodo.org/records/19135682.
(TIFF)

**S32 Fig. Multispecies balancing selection tests in *Mechanitis* at *ivory* and *optix* (left to right).** These analyses included *Hypothyris anastasia*, *Hypothyris euclea*, *Hypothyris ninonia*, and *Hypothyris semifulva*. The red-shaded regions indicate the location of the GWA peaks in the genes *ivory* and *optix*. **(A)** HKAtrans statistics computed in 1 kb windows with a 500 bp sliding step. *Hyalyris antea* was used to determine the ancestral state. Positive HKAtrans values indicate an excess of shared polymorphisms relative to divergence, which is consistent with ancient *trans*-species balancing selection. **(B)** NCD2$_{trans}$ statistics computed in 1 kb windows with a 500 bp sliding step. This statistic measures allele frequency deviations from neutral expectations across multiple species. Values closer to 0 suggest ancient *trans*-species balancing selection. **(C)** NCD2$_{trans\text{-}opt}$ statistics computed in 1 kb windows with a 500 bp sliding step. This statistic is a variant of NCD2$_{trans}$ that optimizes the target frequency at neutrality. Values closer to 0 suggest ancient *trans*-species balancing selection. **(D)** NCD2$_{trans\text{-}sub}$ statistics computed in 1 kb windows with a 500 bp sliding step. A variant of NCD2$_{trans}$ that treats substitutions and polymorphisms separately. Values closer to 0 suggest ancient *trans*-species balancing selection. **(E)** Multispecies nucleotide diversity (π) computed in 10 kb windows. Larger values are indicative of ancient *trans*-species

balancing selection. **(F)** Number of transpolymorphic sites computed in 10 kb windows. Larger values are indicative of ancient *trans*-species balancing selection. The underlying data can be found at https://zenodo.org/records/19135682. (TIFF)

**S33 Fig. *Mechanitis messenoides* CRISPR *ivory* and *optix* mutants.** Mutant phenotypes can be recognized through their asymmetry (wildtype individuals have symmetric left and right wings. **(A)** Wildtype individuals of the *messenoides* and *deceptus* subspecies. **(B)** *ivory* mutants in which orange and black scales have turned yellow. **(C)** *optix* mutants in which orange scales have turned black. (TIF)

**S34 Fig. *Mechanitis messenoides* in situ hybridization for *ivory*.** Comparing nonyellow-banded *Mec. messenoides deceptus* and yellow banded *Mec. messenoides messenoides* shows absence of *ivory* RNA in the yellow band region of the forewing. The colored dots indicate vein-based wing landmarks, demarcating the yellow band region which lacks *ivory* expression in *Mec. messenoides messenoides*. In the nonyellow-banded *Mec. messenoides deceptus*, the red dotted line demarcates the approximate region corresponding to the yellow band region. (TIFF)

**S35 Fig. *Cortex* expression is not associated with the forewing yellow band phenotype.** *Cortex* is expressed across the entire forewings of early fifth instar larvae. Each row shows a set of photos taken on a confocal microscope, with each column showing a different channel (column 1: nuclear DAPI staining (405 nm), column 2: wheat germ agglutinin (WGA) that stains the nuclear membrane (488 nm) or fibrillin, which stains the nucleolus: column 3: *Cortex* antibody (555 nm); column 4: an overlay image with all three channels combined). The last column shows the adult phenotype. Row 1–3: *Mechanitis messenoides deceptus*, with row 2 being a more zoomed image of row 1. Row 4–5: *Melinaea mothone mothone* (no yellow band). Row 6–7: *Melinaea menophilus zaneka* (with yellow band). (TIFF)

**S36 Fig. Differential gene expression analysis in pupal wing discs. (A)** Normalized *ivory* expression levels in the forewing and hindwing across subspecies of *Mechanitis messenoides* (top) and *Melinaea menophilus* (bottom). **(B)** Volcano plots of genome-wide differential expression analysis comparing yellow-banded versus nonyellow-banded forewings in *Mechanitis messenoides* (top) and *Melinaea menophilus* (bottom). Red dots indicate significantly differentially expressed genes; the orange dot corresponds to *ivory*. The underlying data can be found at https://zenodo.org/records/19135682. (TIFF)

**S37 Fig. Transcription factor binding site analysis in *Mechantis messenoides*. (A)** The GWAS peak region ranging to one (weakly- or non-associated) SNP either side of the fixed and most highly associated SNPs (SUPER_6:6877302–6878798) annotated with the motifs identified by homer (yellow arrows) which appear in 100% of sequences for one form and less than 5% of sequences for the other. Highly variable or repetitive motifs which occurred less than 3 times per sequence were excluded. Motifs identified by FIMO[110] which fall within 50 bp of an associated SNP detected by GWAS are shown in blue. Fixed SNPs are highlighted in red, whilst associated SNPs which are not completely fixed are shown in yellow. **(B)** Regions surrounding the homer motifs and closest FIMO motifs are shown in detail. The nucleotide sequences are the consensus sequence from the nonyellow-banded form (*deceptus*, top) and yellow-banded reference form (*messenoides*, bottom). Homer[109] motifs which are enriched in the nonyellow-banded sequences are shown above this consensus sequence, whilst motifs enriched in the yellow-banded form are shown below. FIMO motifs were detected in both sets of sequences. Faded motif labels indicate transcription factors that are not expressed in forewing pupal wing discs. Associated/fixed SNPs are highlighted in the nonyellow-banded sequence according to the nucleotide present. The names of the FIMO motifs correspond to the JASPAR database, whilst the names of the

homer motifs come from the "best-guess" matching motif identified by homer. The underlying data can be found at https://zenodo.org/records/19135682.
(TIF)

**S38 Fig. *Mechanitis messenoides* GWA around *optix* for hindwing black vs orange using categorical grouping versus Patternize scoring for phenotyping. (A)** GWA using phenotype values based on manual, categorical grouping. Individuals were classed as either 0 (semi-melanised, "tiger-patterned" hindwing) or 1 (fully melanic, black hindwing). **(B)** GWA using phenotype values generated through a quantitative, color pattern analysis based approach using Patternize[54]. SNPs above the Bonferroni-corrected significance threshold (dashed orange line) are colored orange. The underlying data can be found at https://zenodo.org/records/19135682.
(TIFF)

**S1 Table. Sample metadata, accessions, and phenotypes.**
(XLSX)

**S2 Table. Genomic locations of GWAS peaks.**
(XLSX)

**S3 Table. *Mechanitis messenoides* wing phenotype scores.**
(XLSX)

**S4 Table. Introgression test details.**
(XLSX)

**S5 Table. Details of f4 tests of introgression.**
(XLSX)

**S6 Table. Details of balancing selection tests.**
(XLSX)

**S7 Table. CRISPR sgRNA sequences used in *Mechanitis messenoides*.**
(XLSX)

## Acknowledgments

We thank Mathieu Joron and Stephen Montgomery for providing samples, Ismael Aldas for support with fieldwork, Nicol Rueda for help with the Relate analysis and Chris Thomas and Jane Hill for comments on the manuscript. ONT library preparation and sequencing were carried out by the Genomics Laboratory at the Bioscience Technology Facility (University of York). The University of York Viking2 HPC and the Wellcome Sanger farm HPC were used for bioinformatics analyses.

## Author contributions

**Conceptualization:** Neil Rosser, Nicola J. Nadeau, Kanchon K. Dasmahapatra, Joana I. Meier.

**Data curation:** Yacine Ben Chehida.

**Formal analysis:** Yacine Ben Chehida, Eva S. M. van der Heijden, Edward Page, Neil Rosser.

**Funding acquisition:** Camilo Salazar, Marianne Elias, Caroline N. Bacquet, Nicola J. Nadeau, Kanchon K. Dasmahapatra, Joana I. Meier.

**Investigation:** Yacine Ben Chehida, Eva S. M. van der Heijden, Edward Page, Patricio A. Salazar C., Kimberly Gabriela Gavilanes Córdova, Mónica Sánchez-Prado, María José Sánchez-Carvajal, Franz Chandi, Alex P. Arias-Cruz, Maya Radford, Kanchon K. Dasmahapatra.

**Methodology:** Yacine Ben Chehida, Eva S. M. van der Heijden, Edward Page, Kanchon K. Dasmahapatra, Joana I. Meier.

**Project administration:** Kanchon K. Dasmahapatra, Joana I. Meier.

**Resources:** Patricio A. Salazar C., Neil Rosser, Gerardo Lamas, Chris Jiggins, James Mallet, Melanie McClure, Camilo Salazar, Marianne Elias, Caroline N. Bacquet, Kanchon K. Dasmahapatra.

**Software:** Yacine Ben Chehida.

**Supervision:** Nicola J. Nadeau, Kanchon K. Dasmahapatra, Joana I. Meier.

**Visualization:** Yacine Ben Chehida, Eva S. M. van der Heijden, Edward Page, Kanchon K. Dasmahapatra, Joana I. Meier.

**Writing – original draft:** Yacine Ben Chehida, Eva S. M. van der Heijden, Edward Page, Nicola J. Nadeau, Kanchon K. Dasmahapatra, Joana I. Meier.

**Writing – review & editing:** Yacine Ben Chehida, Eva S. M. van der Heijden, Edward Page, Patricio A. Salazar C., Neil Rosser, Kimberly Gabriela Gavilanes Córdova, Mónica Sánchez-Prado, María José Sánchez-Carvajal, Franz Chandi, Alex P. Arias-Cruz, Maya Radford, Gerardo Lamas, Chris Jiggins, James Mallet, Melanie McClure, Camilo Salazar, Marianne Elias, Caroline N. Bacquet, Nicola J. Nadeau, Kanchon K. Dasmahapatra, Joana I. Meier.

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
