## [Editor Report · Decision Letter 0]

6 Jan 2026

Dear Dr Dasmahapatra,

Thank you for submitting your manuscript entitled "Genetic parallelism underpins convergent mimicry coloration in Lepidoptera across 120 million years of evolution" for consideration as a Research Article by PLOS Biology. Please accept my sincere apologies for the long delay in getting back to you with feedback due to the closure of the editorial office over the recent holiday period.

Your manuscript has now been evaluated by the PLOS Biology editorial staff, and I am writing to let you know that we would like to send your submission out for external peer review.

IMPORTANT - After discussions within the team, we think that your manuscript would be a better fit as a Short Report at the journal (https://journals.plos.org/plosbiology/s/what-we-publish#loc-short-reports). Upon resubmission (see guidelines below), I would be grateful if you could please tick 'Short Report' as the article type in the dropdown menu in the online submission form.

Before we can send your manuscript to reviewers, we need you to complete your submission by providing the metadata that is required for full assessment. To this end, please login to Editorial Manager where you will find the paper in the 'Submissions Needing Revisions' folder on your homepage. Please click 'Revise Submission' from the Action Links and complete all additional questions in the submission questionnaire.

Once your full submission is complete, your paper will undergo a series of checks in preparation for peer review. After your manuscript has passed the checks it will be sent out for review. To provide the metadata for your submission, please Login to Editorial Manager (https://www.editorialmanager.com/pbiology) within two working days, i.e. by Jan 08 2026 11:59PM.

Kind regards,

Richard

Richard Hodge, PhD

rhodge@plos.org

PLOS

---

## [Decision Letter · Decision Letter 1]

14 Feb 2026

Dear Dr Dasmahapatra,

Thank you for your patience while your manuscript "Genetic parallelism underpins convergent mimicry coloration in Lepidoptera across 120 million years of evolution" went through peer-review at PLOS Biology as a Short Report. Please accept my sincere apologies for the delays that you have experienced during the peer review process. Your manuscript has now been evaluated by the PLOS Biology editors, an Academic Editor with relevant expertise, and by three independent reviewers.

As you will see in the reviewer reports at the end of this e-mail, the reviewers are very positive about your study and agree that it is interesting and well done. In light of the reviews, we are pleased to offer you the opportunity to address the comments from the reviewers in a revision that we anticipate should not take you very long. We will then assess your revised manuscript and your response to the reviewers' comments with our Academic Editor aiming to avoid further rounds of peer-review, although we might need to consult with the reviewers, depending on the nature of the revisions.

In addition, I would be grateful if you could please address the following editorial and data-related requests that I have provided below (A-D):

(A) You may be aware of the PLOS Data Policy, which requires that all data be made available without restriction: http://journals.plos.org/plosbiology/s/data-availability. For more information, please also see this editorial: http://dx.doi.org/10.1371/journal.pbio.1001797

-Supplementary files (e.g., excel). Please ensure that all data files are uploaded as 'Supporting Information' and are invariably referred to (in the manuscript, figure legends, and the Description field when uploading your files) using the following format verbatim: S1 Data, S2 Data, etc. Multiple panels of a single or even several figures can be included as multiple sheets in one excel file that is saved using exactly the following convention: S1_Data.xlsx (using an underscore).

-Deposition in a publicly available repository. Please also provide the accession code or a reviewer link so that we may view your data before publication.

Figure 1B, 2A-B, 3, 4A, S2, S3, S4, S5, S6, S7, S8, S9, S10, S11, S12, S13, S14, S15, S16A, S17C-F, S18C-F, S19C-F, S20C-E, S21C-F, S22C-D, S23C-D, S25, S29A-B, S30A-F, S31A-F, S32A-F, S36A-B, S38A-B

(B) Please also ensure that each of the relevant figure legends in your manuscript include information on *WHERE THE UNDERLYING DATA CAN BE FOUND*, and ensure your supplemental data file/s has a legend.

(C) Please note that we cannot accept sole deposition of code in GitHub, as this could be changed after publication. However, you can archive this version of your publicly available GitHub code to Zenodo. Once you do this, it will generate a DOI number, which you will need to provide in the Data Accessibility Statement (you are welcome to also provide the GitHub access information). See the process for doing this here: https://docs.github.com/en/repositories/archiving-a-github-repository/referencing-and-citing-content

(D) Please ensure that your Data Statement in the submission system accurately describes where your data can be found and is in final format, as it will be published as written there.

**IMPORTANT - SUBMITTING YOUR REVISION**

*Resubmission Checklist*

*Published Peer Review*

*PLOS Data Policy*

*Blot and Gel Data Policy*

Best regards,

Richard

Richard Hodge, PhD

rhodge@plos.org

REVIEWS:

Reviewer #1: This is a beautiful study examining the genetic basis of convergent wing color patterns in mimicry rings. The authors use a rigorous combination of genome sequencing, sophisticated genomic analyses, QTL mapping, CRISPR genome editing, and gene expression studies to identify the genetic basis of color polymorphisms in eight species. Remarkably, the authors find that the same two genes (optix and ivory) underlie convergent color patterns in species that have diverged over 120 million years ago. These results contrast with recent meta-analyses, which have shown that the same genes are used in convergent phenotypes when species are closely related, but not in more distantly related species. Furthermore, the authors show that even in closely related species, gene reuse is not due to allele sharing resulting from hybridization. This result is also in contrast to recent studies in other systems. Together, these results suggest that there are genetic constraints that limit the genes that can be involved in the evolution of color patterns in these mimicry rings. I think this will quickly be a classic paper in the literature on the genetic basis of repeated evolution.

The manuscript is very well-written with clear figures. I just have a few minor points where I found the writing or figures were not completely clear. Also, a note to the authors to please use page and line numbers to facilitate comments!

1. In the second paragraph of the main text, the sentence: "Where genes are reused, convergence may result from independent evolution of the same phenotype…" is unclear. I think you mean "convergence may result from independent mutation at the same gene". Otherwise, you are just defining convergence.

2. In the same paragraph, reference 10 is used to make the point that the same phenotypes can evolve from ancestral standing variation. I think that Colosimo et al. 2005 Science would be a better reference, as it specifically is focused on the genetics of repeated phenotypic evolution and the Jones et al. 2012 Nature paper is focused on repeated adaptation.

3. In the paragraph on the forewing yellow band in Ithomiini butterflies, it would be helpful to provide more information about what the E230 cis-regulatory element is and how it relates to the ivory micro-RNA.

4. In the first paragraph on hindwing melanisation in Ithomiini butterflies, it is mentioned that additional associated SNPs in some species are likely due to population structure. But, is there a way to deal with this confounding factor in the association mapping analyses?

5. In the last sentence of the paragraph on "Genetic architecture of colour patterning in the moth…:, I would move the phrase "between lineages that diverged ~120 MYA" to the very end of the sentence.

6. In the first paragraph of the section "Repeatable and predictable evolution", I don't really see the evidence in the paper for "modifiers located next to these switches…". Indeed, only two loci of major effect are identified, but what is the evidence for linked modifiers?

7. In the second paragraph of the same section, the first sentence is a bit convoluted. Perhaps it should be broken into two sentences to more clearly convey the punchline.

8. In this same paragraph, the authors claim that the limited number of paths might enable diverse taxa to join the mimicry ring more easily. But, I could argue that the limited and narrow pathway makes it very difficult! The key here is likely waiting time for mutations in these genes. In the future, it would be very interesting to know whether mutation rates at these loci are particularly high relative to the rest of the genome, as in the stickleback Pitx1 locus (Xie et al. 2019 Science).

9. Figure 2: why are there two regions of strong association at optix in Mel. marsaeus? Is this possibly an assembly error? A brief comment on this result would be welcome. Also, in the Figure 2 legend, please define ivory pro.

10. Figure 4: I would appreciate pictures of the wild-type wings for comparison with the knockouts, especially for ivory, where the regions in which the gene have been deleted are not denoted. Also, in the legend for panel C, please indicate that the ivory expression is in white on the photo.

Reviewer #2: This manuscript documents convergent evolution of the genetic basis of convergent color patterns across highly divergent members of lepidopteran mimicry rings. The authors use genome-wide association studies to identify loci associated with color pattern in eight species, and find that the significant loci consistently include the genes ivory and optix. This suggests that the genetic mechanisms for transitions between the color phenotypes observed in this mimicry ring are highly constrained. The authors further demonstrate that even among closely related species, the shared genetic basis of color pattern is typically due to independent reuse of the same genes rather than introgression of the same mutations, and use Crispr-Cas9 knockouts and in situ hybridization to confirm the causal role of ivory and optix in one species.

This study is likely to be of broad interest due to the surprisingly high level of convergence in the genetics of color pattern. The authors are admirably thorough in addressing follow-up questions to the main result, the writing is clear and concise, and the methods and interpretations of results are appropriate. My only suggestion for improvement is to consider placing this study in a broader context of gene reuse, by noting other groups of organisms and other types of traits where similar patterns have been observed. In particular, color differences between and within vertebrate species, especially birds, often map to the same small set of genes. This might suggest that developmental pathways underlying coloration are more constrained than other traits frequently involved in local adaptation and/or species differences.

Reviewer #3: This study addresses a deep and central question in evolutionary biology: What is the molecular basis of convergent phenotypes, and through what mechanisms do new forms arise?

A major novelty of this study is that it encompasses a very wide range of evolutionary timescales (1-120 Mya) to study the evolutionary genetic basis of convergent wing pattern polymorphisms. The striking finding is that alleles at the same two loci (ivory and optix) control equivalent wing pattern components across highly diverged species, including butterflies and a moth that form part of a complex of mimicry rings. The scale of the study is also impressive, requiring comparative whole genome analysis of multiple wing-pattern forms of seven species' population samples collected in the field. Functional validation of these loci is demonstrated for one of the species using CRISPR-Cas9 mosaic KO and in-situ hybridisation of developing wings.

Of particular interest and novelty is the discovery of a large inversion polymorphism in the moth (Chetone histrio) whose breakpoint junctions closely parallel a classic inversion polymorphism in Heliconius numata that is presumed to function as a supergene.

The other novelty of the study is that it finds no evidence that these alleles have been transferred horizontally via occasional inter-species hybridisation events, implying that they have arisen through de novo mutation many times. This suggests that the contrasting pattern observed within Heliconius may be restricted to closely related species.

The interpretation is appropriately cautious, recognising some limitations and outstanding questions.

I only have very minor presentational suggestions.

Main text, para 3: "..single butterfly species or genus" is followed by citations and sentences referring to two different genera

Main text, "Repeatable and predictable evolution" section: semi-colon after "dispute"; the first sentence in the following paragraph doesn't quite make sense

Fig. 2. As the first narrative section relating to this figure is restricted to Ithomiini, would be useful to make it obvious which these are in the figure (e.g. vertical line)

Fig. 3. Light-blue shaded blocks would give greater visibility to significance colours and avoid confusion with the second use of purple

---

## [Editor Report · Decision Letter 2]

20 Mar 2026

Dear Kanchon,

On behalf of my colleagues and the Academic Editor, Leonie Moyle, I am pleased to say that we can in principle accept your manuscript for publication, provided you address any remaining formatting and reporting issues. These will be detailed in an email you should receive within 2-3 business days from our colleagues in the journal operations team; no action is required from you until then. Please note that we will not be able to formally accept your manuscript and schedule it for publication until you have completed any requested changes.

PRESS

Best wishes,

Richard

Richard Hodge, PhD

rhodge@plos.org

PLOS
